## PROCEEDINGS A

atmospheric chemistry, biogeochemistry, oceanography

ocean acidification, marine trace gases, climate, atmospheric chemistry

**Author for correspondence:**
Frances E. Hopkins
e-mail: fhop@pml.ac.uk

# The impacts of ocean acidification on marine trace gases and the implications for atmospheric chemistry and climate

Frances E. Hopkins[1], Parvadha Suntharalingam[2], Marion Gehlen[3], Oliver Andrews[4], Stephen D. Archer[5], Laurent Bopp[6,7], Erik Buitenhuis[2], Isabelle Dadou[8], Robert Duce[9,10], Nadine Goris[11], Tim Jickells[2], Martin Johnson[2], Fiona Keng[12,13], Cliff S. Law[14,15], Kitack Lee[16], Peter S. Liss[2], Martine Lizotte[17], Gillian Malin[2], J. Colin Murrell[2], Hema Naik[18], Andrew P. Rees[1], Jörg Schwinger[11] and Philip Williamson[2]

[1]Plymouth Marine Laboratory, Prospect Place, Plymouth, UK
[2]School of Environmental Sciences, University of East Anglia, Norwich Research Park, Norwich NR4 7TJ, UK
[3]Laboratoire des Sciences du Climat et de l'Environnement, Institut Pierre Simon Laplace, Orme des Merisiers, Gif-sur-Yvette cedex, France
[4]School of Geographical Sciences, University of Bristol, University Road, Bristol BS8 1SS, UK
[5]Bigelow Laboratory for Ocean Sciences, East Boothbay, ME, USA
[6]Laboratoire de Météorologie Dynamique, Institut Pierre-Simon Laplace, CNRS-ENS-UPMC-X, Département de Géosciences, Ecole Normale Supérieure, France
[7]Université Ecole Polytechnique, Sorbonne Université, Paris, France
[8]Laboratoire d'Etudes en Géophysique et Oceanographie Spatiales, University of Toulouse, Toulouse, France
[9]Department of Oceanography, and [10]Department of Atmospheric Sciences, Texas A&M University, College Station, TX, USA
[11]NORCE Climate, Bjerknes Centre for Climate Research, Bergen, Norway

[12]Institute of Ocean and Earth Sciences (IOES), University of Malaya, Kuala Lumpur, Malaysia

[13]Institute of Graduate Studies (IGS), University of Malaya, Kuala Lumpur, Malaysia

[14]National Institute of Water and Atmospheric Research, Wellington, New Zealand

[15]Department of Chemistry, University of Otago, Dunedin, New Zealand

[16]Division of Environmental Science and Engineering, Pohang University of Science and Technology, Pohang, South Korea

[17]Department of Biology, Université Laval, Quebec City, Canada

[18]CSIR-National Institute of Oceanography, Dona Paula 403004, Goa, India

FEH, 0000-0002-2991-5955; LB, 0000-0003-4732-4953

Surface ocean biogeochemistry and photochemistry regulate ocean–atmosphere fluxes of trace gases critical for Earth's atmospheric chemistry and climate. The oceanic processes governing these fluxes are often sensitive to the changes in ocean pH (or $pCO_2$) accompanying ocean acidification (OA), with potential for future climate feedbacks. Here, we review current understanding (from observational, experimental and model studies) on the impact of OA on marine sources of key climate-active trace gases, including dimethyl sulfide (DMS), nitrous oxide ($N_2O$), ammonia and halocarbons. We focus on DMS, for which available information is considerably greater than for other trace gases. We highlight OA-sensitive regions such as polar oceans and upwelling systems, and discuss the combined effect of multiple climate stressors (ocean warming and deoxygenation) on trace gas fluxes. To unravel the biological mechanisms responsible for trace gas production, and to detect adaptation, we propose combining process rate measurements of trace gases with longer term experiments using both model organisms in the laboratory and natural planktonic communities in the field. Future ocean observations of trace gases should be routinely accompanied by measurements of two components of the carbonate system to improve our understanding of how *in situ* carbonate chemistry influences trace gas production. Together, this will lead to improvements in current process model capabilities and more reliable predictions of future global marine trace gas fluxes.

## 1. Introduction

The interface between the ocean and the atmosphere is a crucial boundary of the Earth system. It controls not only the exchange of substances which influence the chemistry of the atmosphere and our climate, but also the transfer of essential elements vital for human health and ecosystem functioning, from the ocean to the land. The Earth system is currently facing unprecedented changes in global biogeochemical and physical processes, driven by human emissions of greenhouse gases [1]. In this review, we focus on one such change, ocean acidification (OA), and assess its impact upon the production of marine trace gases and resulting feedbacks to the atmosphere. We discuss the role of marine trace gases in the Earth system's chemistry and climate, and provide an overview of the state-of-the-knowledge of the marine trace gas response to OA derived from both experimental and modelling studies. In addition, we consider regions especially sensitive to OA, and discuss the effects of other environmental changes, such as rising temperatures and ocean deoxygenation, on the production and emission of marine trace gases.

## 2. Marine trace gases

The surface ocean is a key source of a variety of trace gases, which flux to the atmosphere and play critical roles in the Earth's biogeochemical cycles, and strongly influence the chemistry of its atmosphere and its radiative budget. These include greenhouse gases, such as carbon dioxide ($CO_2$), nitrous oxide ($N_2O$) and methane ($CH_4$), that have relatively well-understood effects on

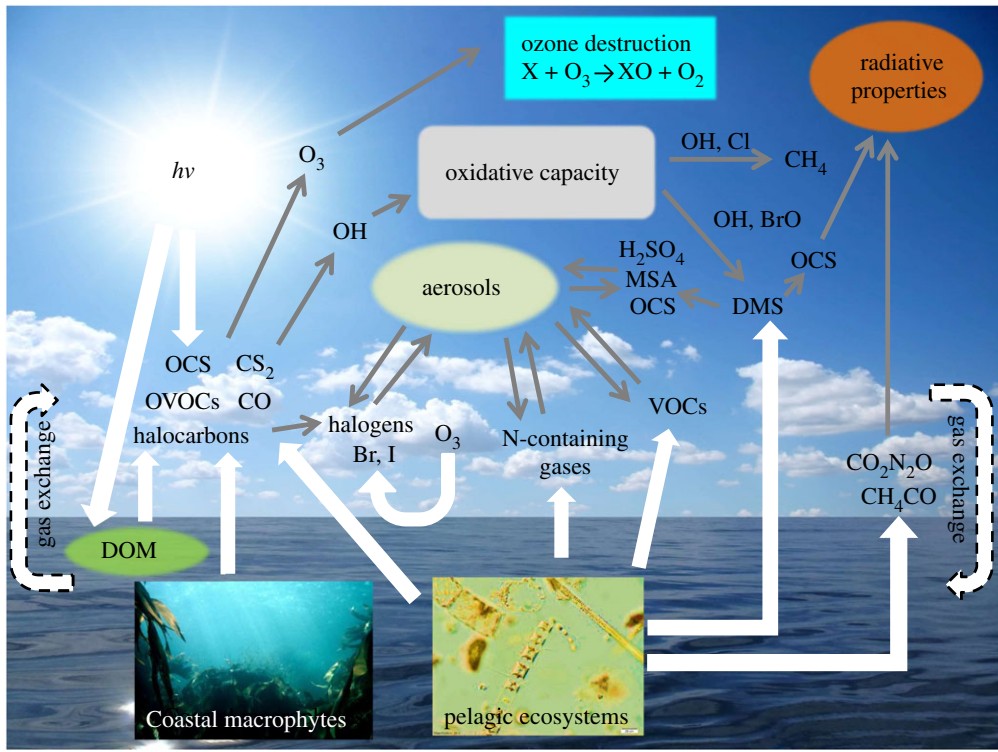

**Figure 1.** Overview of the production of marine trace gases and their roles in atmospheric and climatic processes. (Online version in colour.)

global radiative forcing and atmospheric chemistry [2]. In addition, the ocean releases a range of biogenic volatile organic compounds (BVOCs), containing carbon, sulfur, nitrogen and halogens (figure 1). The transfer of these compounds from ocean to land via the atmosphere represents a key step in the global cycling of essential elements that provide benefits to ecosystem function and human health [3]. Furthermore, the atmospheric oxidation products of some trace gases, such as dimethyl sulfide (DMS), methylamines and a variety of BVOCs, can impact upon marine aerosols, thereby influencing cloud-related processes and global radiative forcing [4–8]. Other marine trace gases, including halocarbons and oxygenated volatile organic compounds (OVOCs), produce highly reactive atmospheric radicals that readily destroy protective stratospheric ozone ($O_3$), and drive the rapid cycling of tropospheric photo-oxidants and $O_3$ with implications for coastal air quality [9–12]. Biogenic marine trace gases are directly produced by micro- and macro-algae, and by prokaryotic microbes [13–15]. They are also released from sediments [16], seafloor seeps [17,18], as a result of bacterial degradation of precursor compounds [19–22], and via reactions between organic matter, sunlight and $O_3$ [23] (figure 1). Whereas the sources and sinks of $CO_2$, $N_2O$ and DMS, are reasonably well established, others remain poorly understood. Given the critical role marine trace gases play in atmospheric chemistry and climate-related processes, it is important to consider the influence of global environmental change on their oceanic flux, and associated feedbacks to climate.

## 3. CO$_2$-driven ocean acidification

Anthropogenic $CO_2$ emissions from burning fossil fuels and land-use change are currently the primary driver of global climate change [24]. Atmospheric $CO_2$ concentrations have steadily risen over the last 150 years and are now higher than at any time during at least the last 800 000 years of Earth's history [25,26]. This rise directly results in increased oceanic $CO_2$ absorption

[27] and OA. In addition to increased hydrogen ion concentration ($H^+$), and hence decreased pH, associated chemical changes include increased concentrations of bicarbonate ions ($HCO_3^-$) and reductions in carbonate ions ($CO_3^{2-}$) [27,28]. Globally, a decrease in surface ocean pH of approximately 0.1 units has already occurred relative to preindustrial times, with a projected fall of a further approximately 0.3 units by 2100 under high-emission scenarios [1,29,30]. Sustained ocean observations from seven globally distributed time-series stations, including the northerly Iceland and Irminger Seas, the subtropical Bermuda Atlantic Time-series Study (BATS) and the tropical Hawaiian Ocean Time-series (HOT) show a 0.013–0.025 pH unit per decade decline since the 1980s [31]. This rate of change to ocean biogeochemistry is rapid on geological timescales and is probably unprecedented in the last 300 million years of Earth's history [32]. Most OA research has focused on potential effects on calcifying organisms (e.g. [33–37]) and other ecologically and economically important species [38–41]. To date, there has been little assessment of OA impacts upon marine biogeochemical cycles that potentially involve changes in the production of marine trace gases and associated feedbacks on atmospheric chemistry and climate [42–45]. Given the documented sensitivity of marine microbes to a variety of environmental factors (e.g. seawater chemistry, temperature; [46]), there is clearly potential for climatically significant future change in the production and emissions of trace gases. Ultimately, OA will act in concert with global warming and modify the physical–chemical environment of pelagic and benthic communities. This in turn is expected to trigger changes in species composition of micro- and macroalgal communities [47–51]. Shifts in the community composition, or any increase in stress arising from OA, may alter the production of trace gases and the geographical pattern of trace gas emissions.

## 4. Biological processes drive trace gas production

We focus on biogenic marine trace gases (excluding $CO_2$) that are directly and/or indirectly produced by bacteria, phytoplankton and seaweeds, as well as trace gases produced by reactions involving dissolved organic matter (DOM). Given the known and predicted effects of OA on biological processes [52], it is likely that the net production of biogenic trace gases (including both production and loss processes) may be influenced by OA. First, we provide a general overview of the potential effects of OA on biological processes related to marine trace gas production, while later we discuss specific trace gases in greater detail.

Marine bacteria drive essential biogeochemical processes, including organic matter decomposition, elemental cycling and nutrient regeneration, and are key players in the production and consumption of marine trace gases [53,54] (figure 1). While we have a reasonable understanding of the influence of temperature and organic matter availability on bacterial processes, the role of OA is less well constrained [46,55]. The majority of studies imply that bacterial processes are seemingly resilient to reductions in pH due to the rapid microbial cell division and the associated adaptive potential that this affords, but our mechanistic understanding of this, and how it may influence trace gas production, is limited [56–58]. Recent work has shown that pH-induced changes in bacterial physiological processes affect cellular energy allocation, thereby influencing fluxes of carbon and energy in microbial systems [59]. In addition, OA may lead to increased exudation of dissolved organic carbon by phytoplankton, altering substrate availability and form [60] and enhancing extracellular enzyme activity [61–63]. Such changes have the potential to indirectly influence trace gas levels by altering the availability of precursors in the dissolved organic carbon pool, and by influencing the rate of bacterial processes that both produce and consume trace gases, and that ultimately result in their net production.

Many marine photosynthesizers, including single-celled phytoplankton and macrophytes such as seaweeds, are a direct source of marine trace gases, including DMS [64] and certain halocarbons, such as methyl iodide ($CH_3$) and bromoform ($CHBr_3$) [65,66]. DMS production by algal cells may be a means to release excess carbon and sulfur via a metabolic overflow mechanism [20,67], or alternatively, it may provide protection against physiological stress [68,69].

Similarly, halocarbon production has been attributed to relief from stressors including light-induced [70], oxidative [71] or mechanical [13] stress. In general, the biosynthesis of trace gases and their precursors is poorly understood, but is likely to vary substantially between species. For example, intracellular concentrations of dimethylsulfoniopropionate (DMSP), the algal precursor for DMS, have been shown to vary by up to three orders of magnitude across phytoplankton groups within a population [72].

Shifts in phytoplankton community composition in response to OA might alter global DMS emissions. For example, coccolithophores, a relatively well studied and ubiquitous group of marine calcifying haptophytes, possess high levels of intracellular DMSP, with blooms of this species associated with the release of vast quantities of DMS [64]. Their fate under OA has been the target of numerous experimental studies and field surveys over the past 20 years. However, the direction and magnitude of the response to OA varies substantially [33,73–75] and is still poorly understood. This underlines our still limited process understanding, which continues to hinder our ability to anticipate changes in DMS emissions under future climate change.

# 5. Experimental evidence: exploring effects of ocean acidification on marine trace gases

Our knowledge of the effects of OA on marine trace gas production stems from the results of a suite of experimental approaches, summarized in table 1. At the simplest level, on an experimental spectrum of complexity, are incubations with single-species algal cultures (less than 1 l, 2–3 replicates and 7–40 days). This is an approach which, given the reduced complexity, serves as a means to establish baseline concepts and identify the most sensitive or relevant physiological processes and mechanisms for trace gas production. Studies have considered ambient $CO_2$ versus one high $CO_2$ treatment: 370–395 µatm compared with 750–1000 µatm, corresponding to pH treatments of 8.1–8.3 (ambient) and 8.0–7.7 (High $CO_2$) (table 1). Of greater complexity and closer to actual ocean conditions are *in situ* mesocosm experiments, essentially giant 'test tubes' that allow large-scale, field-based, community-level assessments of the effects of OA on natural surface ocean communities (2400–75 000 l, 1–3 replicates and 25–35 days). Mesocosms provide an understanding of the net effects on the whole community response to OA, in many cases investigated under conditions of high productivity and growth associated with a phytoplankton bloom. Earlier experiments considered two triplicated $CO_2$ treatments ('Ambient' $CO_2$ 300–400 µatm, pH 8.2–8.1 versus High $CO_2$ 700–900 µatm, pH 7.9–7.8) [61,77,84,87] (table 1). Later experiments considered a wider range of $CO_2$/pH treatments (175–3000 µatm, pH 8.3–7.3) using a gradient of treatments levels across up to nine mesocosm enclosures, and allowing linear relationships between $CO_2$/pH and response parameters to be determined [79,82,83,85,86,88] (table 1). The observed OA effects on trace gases may reflect a combination of stress responses including acclimation, population re-structuring and associated adaptation [89]. Shipboard microcosm experiments are a useful tool to bridge the gap between complex mesocosm experiments and simple culture experiments (5–10 l, 3–12 replicates and 4–10 days). Conducting multiple short-term experiments over extensive spatial scales enables both the physiological effects of OA to be assessed, as well as the spatial variability in responses of surface ocean communities to future OA scenarios. The experimental design, involving relatively small incubation volumes of 5–10 l, allows multiple $CO_2$ treatments to be considered. For 18 experiments performed over a range of temperate and polar waters, Hopkins & Archer [80] and Hopkins *et al*. [90] used four $CO_2$ treatments in triplicate (Mid: 533.4 ± 40.0 µatm, pH 7.9 ± 0.03, High: 673.8 ± 82.2 µatm, pH 7.8 ± 0.1, High+: 841.5 ± 128.2 µatm, pH 7.8 ± 0.1, High++: 1484.0 ± 104.0 µatm, pH 7.5) and an ambient control (320.2 ± 38.3, pH 8.1 ± 0.1). Hussherr *et al*. [81] adopted a different approach by exposing phytoplankton communities within six single incubations to a pH gradient, from 509 µatm (pH 7.94) to 3296 µatm (pH 7.16) (table 1). In the following section, we use information from all three types of experiments to consider the impacts of OA on trace gas production from the cellular to the community level, and in table 2,

we provide an overview of the types of response to OA that may result in changes in trace gas production.

To place the data in the context of key ocean–atmosphere linkages, we group the trace gases according to their primary atmospheric roles: those that primarily influence aerosols and cloud albedo, those that play a key role in regulating oxidative capacity and those that exhibit direct radiative effects, while recognizing that some trace gases possess multiple atmospheric roles. A large proportion of the following discussion focuses on DMS, since the amount of information available for this trace gas currently dwarfs the available information for all others. Furthermore, research into the net production of DMS in the surface oceans has been prominent within the fields of marine biogeochemistry and sea–air interactions for more than three decades due to the global significance of its role in climatic and atmospheric processes. The significance of oceanic DMS production and emission and its potential role in influencing global climate and atmospheric chemistry was highlighted in the 1980s by the seminal publication of Charlson *et al*. [4]; this spurred more than three decades of intensive investigation into the marine biogeochemistry, air–sea interactions and climate impacts of oceanic DMS.

# 6. Atmospheric role: aerosols and albedo

## (a) Dimethyl sulfide and dimethylsulfoniopropionate

DMS is produced via enzymatic breakdown of the algal and bacterial secondary metabolite DMSP [20,93]. The release of intracellular DMSP into the surrounding seawater, and its subsequent and rapid conversion to DMS, is triggered by a number of processes including the active exudation of DMSP from living cells, and cell lysis during senescence, viral attack or grazing [20]. Most of the resulting DMS undergoes rapid processing in seawater via both bacterial (50–88%) and photochemical (8–34%) pathways [94]. The remaining DMS, which amounts to around 4–16% of total production, ventilates from the surface ocean to the lower atmosphere [95].

Upon entering the atmosphere, DMS undergoes rapid oxidation to species including $SO_2$, $H_2SO_4$, methanesulfonic acid (MSA) and dimethylsulfoxide (DMSO), thereby contributing to aerosol formation and growth and to atmospheric acidity [4,96]. When simulated at a global level, annually averaged DMS-derived aerosol radiative forcing at the top of the atmosphere has been estimated to have a climate-cooling effect of between $-1.69\,W\,m^{-2}$ [97] and $-1.79\,W\,m^{-2}$ [98]. For context, greenhouse gas radiative forcing for 1750–2011 was estimated at $+1.83\,W\,m^{-2}$ for $CO_2$, $+0.61\,W\,m^{-2}$ for $CH_4$ and $+0.17\,W\,m^{-2}$ for $N_2O$ [99]. DMS is also a major source of cloud condensation nuclei (CCN) via its rapid gas-phase oxidation to sulfuric acid ($H_2SO_4$), which influences the radiative properties of clouds, both microscopically via cloud droplet number concentration and effective radius, and at the larger scale by influencing cloud abundance, albedo and lifetime [100–102]. DMS also plays a significant role in the atmospheric oxidation pathways of other key trace gases, including isoprene, ammonia and halocarbons [103–105].

The response of DMS to increasing ocean acidity has been studied at several levels of biological and environmental complexity. At a fundamental level, DMS and DMSP concentration changes have been monitored in single-species cultures of algae exposed to relatively short-term (7–25 days) variations in OA. These studies report the response in two seaweed species, *Ulva lactuca* and *U. clathrate,* three strains of the well-studied DMSP-producer *Emiliania huxleyi* and two species of diatom [76–78,88,106]. The results of these studies, each following different experimental approaches, indicate a variety of physiological responses to short-term OA exposure between different species and strains of algae (electronic supplementary material, table S1). Such experiments exclude key bacterially mediated processes and do not consider how the activity of micro- and mesozooplankton (grazers) may be affected by OA. This illustrates the challenge of predicting the response of natural communities on the basis of the response of single species, and emphasizes the need for more in-depth understanding of the physiological roles of DMSP and DMS.

**Table 1.** Overview of experimental methods employed in studies of the effects of OA on marine trace gases.

| Experimental Technique | Vol. (l) | Number of replicates | Duration (days) | Key studies | CO₂ and pH treatments CO₂ (μatm) | pH | Method of acidification | What can it tell us? | Strengths | Weaknesses |
|---|---|---|---|---|---|---|---|---|---|---|
| Single-species algal cultures | <1 | 2–3 | 7–40 | [76–79] | 385/1000 370/760 Ambient/790 395/900 | 8.1/7.7 no data 8.3/8.0 8.1/7.7 | aeration with $CO_2$-enriched air, pH-stat to maintain constant DIC and pH | Batch cultures: acclimated/ physiological response to OA  Semi-continuous culture: multiple generations allow insight into adaptive plasticity to OA  Level of sensitivity to OA/high $CO_2$ | Useful tool for establishing baseline concepts  Reduced complexity compared with natural populations  Determines direct response on trace gas production by phytoplankton isolates (if axenic)  High duplication/ reproducibility | Do not simulate complex natural systems  Elimination of extracellular (bacterial) processes that may be key control on trace gas production |
| Shipboard microcosm experiments | 5–10 | up to 12 | 4–10 | [80] [91] [81] | Av. of 18 expts: $320.2 \pm 38.3$ $533.4 \pm 40.0$ $673.8 \pm 82.2$ $841.5 \pm 128.2$ $1484.0 \pm 104.0$  5 treatments and control over range: 509–3296 | $8.1 \pm 0.1$ $7.9 \pm 0.03$ $7.8 \pm 0.1$ $7.8 \pm 0.1$ 7.5  7.9–7.2 | addition of strong acid/base, e.g. $HCl/NaHCO_3^-$ | Physiological response and extent of the variability in response/plasticity between communities  Level of sensitivity to OA/high $CO_2$ | Extensive spatial coverage  Natural gradients in carbonate chemistry, temperature, nutrients  Multiple short-term identical experiments on complex natural communities  Results in large, highly replicated, statistically robust data sets | Short-term physiological response: representative?  Bottle effects  Rapid acidification |

(*Continued.*)

**Table 1.** (*Continued.*)

| Experimental Technique | Vol. (l) | Number of replicates | Duration (days) | Key studies | CO₂ and pH treatments | | Method of acidification | What can it tell us? | Strengths | Weaknesses |
|---|---|---|---|---|---|---|---|---|---|---|
| | | | | | $CO_2$ (µatm) | pH | | | | |
| mesocosm experiments | 2400–75 000 | 1–3 | 25–35 | [82] | 175–1085 | 8.3–7.6 | aeration with $CO_2$-enriched air, or addition of $CO_2$-saturated seawater | Whole community response during bloom conditions | Close to natural conditions (light and temperature) + large volume | Limited by number of experimental replicates |
| | | | | [83] | 400–1252 | 8.1–7.6 | | | | Difficult to test multiple drivers |
| | | | | [77] | ambient versus 700 | 8.2 versus 7.8 | | Acclimation (>30 days) | Longer timescale = improved realism of representation of surface ocean | |
| | | | | [84] | 300 versus 780 | 8.1 versus 7.8 | | Net production by whole community and associated biogeochemistry | | Logistically challenging (physically and financially) |
| | | | | [85] | 175–1085 | 8.3–7.6 | | | Towards a whole community, adaptive response | Minimal geographical coverage |
| | | | | [61] | 400 versus 900 | no data | | | | |
| | | | | [86] | 160–830 | no data | | | | |
| | | | | [87] | 350 versus 700 | 8.1 versus 7.9 | | | | |
| | | | | [79] | 280–3000 | 8.1–7.3 | | | | |
| | | | | [88] | 330–1166 | 7.9–7.5 | | | | |

**Table 2.** Overview of types of response to OA relevant to trace gas production and cycling.

| type of response to ocean acidification | description/example | Relevance to which trace gases? |
|---|---|---|
| direct chemical | effect of OA on chemical processes/equilibria that regulate trace gas production<br>e.g. pH-induced shift from $NH_3$ to $NH_4^+$ leads to reduced $NH_3$ emissions [91] | $NH_3$<br>methyl amines |
| direct biogeochemical | effect of OA on biogeochemical processes that regulate trace gas production<br>e.g. pH-induced reduction in $NH_3$ leads to reduced nitrification and reduced $N_2O$ production [92] | $N_2O$<br>$NH_3$ |
| direct biological | effect of OA on organism-level processes that regulate trace gas production<br>e.g. pH-induced reduction in calcification in coccolithophores, leads to reduced abundance and reduced DMS emissions [79] | DMS(P)<br>halocarbons<br>CO<br>isoprene |
| indirect biological | effect of OA on availability/type of organic substrates that regulate trace gas production<br>e.g. pH-induced increase in DOC exudation by phytoplankton enhances substrate/precursor availability [60] which may affect trace gas production | halocarbons<br>OVOCs<br>CO<br>OCS, $CS_2$<br>isoprene |
| community level | effect of OA on community-level processes/community structure that regulate trace gas production and cycling<br>e.g. high $CO_2$(aq)-induced community-level shift towards dinoflagellates with low $CO_2$(aq) affinity and increased DMS(P) producing ability [82] | DMS(P)<br>halocarbons<br>$N_2O$<br>OVOCs<br>CO<br>OCS, $CS_2$<br>isoprene |

At a more complex community level, the DMS response to OA has also been assessed using shipboard microcosms [80,81,90] (table 1). Such experiments test the resilience of natural communities to abrupt manipulation of carbonate chemistry, generally in controlled conditions using simulated light and temperature regimes. Of the 18 microcosm experiments reported in the meta-analysis by Hopkins *et al.* [90], all 11 experiments performed in the temperate waters of the northwest European shelf showed consistent large and significant increases in DMS in response to OA. By contrast, the seven experiments carried out in Arctic and Southern Ocean waters exhibited minimal OA influence on net DMS production. The discrepancy in DMS response between shipboard microcosms in temperate versus polar waters was hypothesized to be a product of variable levels of sensitivity of the respective communities to changes in the mean state of

carbonate chemistry [107]. In comparison to the well-buffered temperate waters of the northwest European shelf, polar waters are poorly buffered with respect to the addition of $CO_2$, resulting in a naturally large variability in carbonate chemistry [108]. Furthermore, the polar oceans experience unique biogeochemical processes, such as sea-ice formation and melt, iron-stimulated ice-edge blooms and under-ice organic matter respiration that also contribute to large natural variability in carbonate chemistry [109–111]. Thus, relative to the temperate communities, polar communities may be adapted to, and may be able to tolerate, large variations in carbonate chemistry, as reflected in the low sensitivity of DMS production to OA in polar waters. Of course, this hypothesis may not be universally applicable. A further 9 day microcosm study in polar waters performed during the summer [81] illustrated a substantial decrease in DMS concentrations with increased $CO_2$ and a less substantial but significant decrease in particulate DMSP concentrations. Such contrasting results may be unsurprising, given that the complexity of the DMS response to OA and the influence of a multitude of factors. For example, beyond the latitudinal variability in carbonate chemistry discussed in Richier *et al*. [107], the Arctic Ocean itself possesses high regional carbonate chemistry variability [112], related to sea-ice formation and the input of riverine and meltwater. Furthermore, spatial or seasonal differences in phytoplankton community composition, as well as the associated variability in physiological response (e.g. DMSP synthesis), could result in contrasting DMS responses to OA [77,80]. Alternatively, the overall DMS response may be associated with distinctive impacts on the transformation of DMSP via zooplankton grazing [61,86] or bacterial activity [79,81,82]. Given the wide variability in plankton community composition, activity and turnover rates, this emphasizes the need to consider both spatial and seasonal contexts when evaluating the sensitivity of DMS production to OA.

The majority of studies on the influence of OA on DMS have considered the response in large mesocosm enclosures (approx. 24–55 $m^3$), incubated under close to natural environmental conditions. Such experiments incorporate the response of complex natural planktonic communities and the multiple processes that control the concentration of DMS in surface waters. Figure 2 provides an overview of nine mesocosm experiments carried out by different research groups in five Northern Hemisphere locations, ranging from arctic to subtropical latitudes and covering early summer to winter seasons. When DMS concentrations are integrated over the duration of each experiment, the difference in concentration between $p$$CO_2$ treatments of approximately 350 µatm versus approximately 750 µatm varied from +26% to −42%, with seven of the nine experiments showing decreased DMS concentrations with increased acidity (figure 2; electronic supplementary material, table S2). These $p$$CO_2$ levels approximate, respectively, the average ambient conditions at the time of experiments and the twofold increase that could occur by 2100 according to the Intergovernmental Panel on Climate Change (IPCC) Representative Concentration Pathways RCP6.0 emissions scenario [113].

Predominantly, a decrease in concentrations of DMS in response to OA in mesocosm experiments has been observed, which suggests that a dominant control on DMS net production is affected by a change in carbonate chemistry or $H^+$ concentration. DMSP production by phytoplankton is highly species-specific and several studies have demonstrated correlations between phytoplankton community composition and DMSP concentrations that may have influenced the response of DMS production to OA [77,79,82,83,86] (table 2). On several occasions, specific aspects of DMS production have been examined: direct measurements of the rates of DMSP synthesis [82], grazing rates on phytoplankton that may enhance conversion of DMSP to DMS [61,86], the *in vitro* activity of DMSP-lyase enzymes that convert DMSP to DMS [86] and rates of bacterial metabolism of DMSP and conversion to DMS [83]. Nonetheless, disentangling the complex processes responsible for the observed changes in DMS remains challenging.

The variety of ways in which data from the nine large-scale mesocosm experiments have been interpreted also creates a further challenge when attempting to unify the DMS response to OA. Time-integration of DMS concentrations across the full duration of each experiment (figure 2) is a start towards consistent interpretation but a more in-depth approach is required to fully compare experiments and extract a comprehensive evaluation. As an example, a large significant

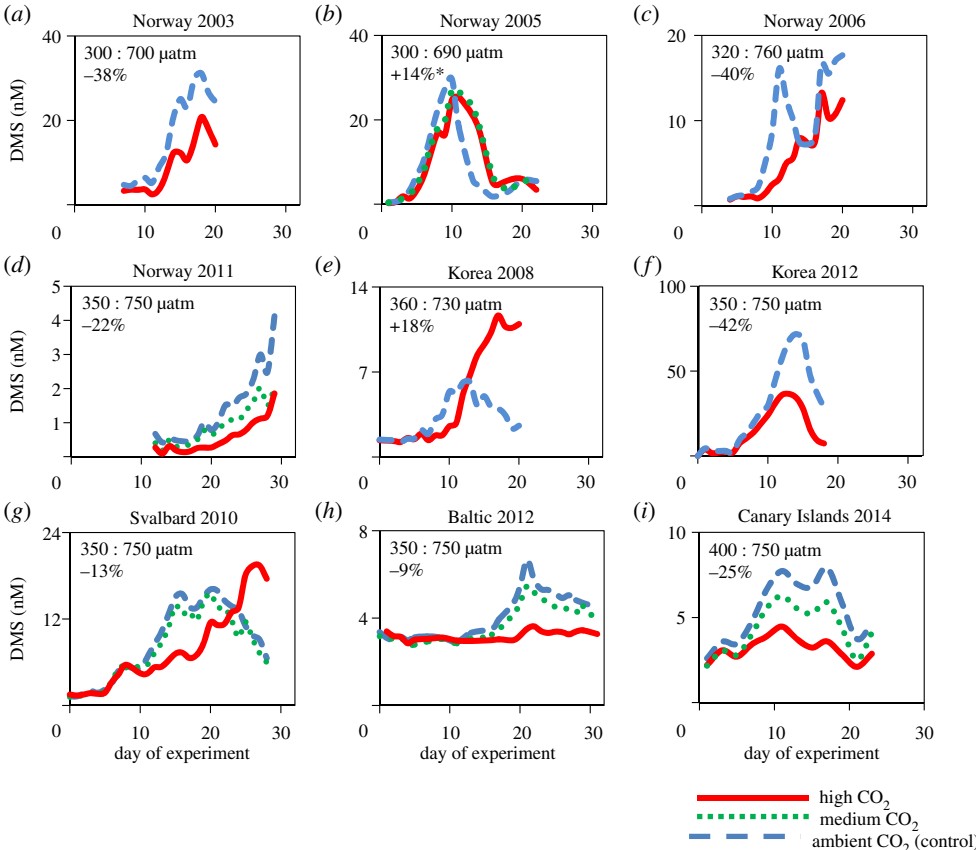

**Figure 2.** Overview of the DMS response from all published OA mesocosm experiments carried out under natural environmental conditions, to date. Four experiments took place in early summer in Raunefjord, Norway (60.3°N, 5.2°E): (*a*) Avgoustidi *et al*. [77], (*b*) Vogt *et al*. [87], (*c*) Hopkins *et al*. [84], (*d*) Webb *et al*. [79]; two in the coastal waters of Jangmok, Korea (34.6°N, 128.5°E): one in winter (*e*) Kim *et al*. [61] and the other in early summer (*f*) Park *et al*. [86]; single experiments were carried out in (*g*) summer in the Svalbard Archipelago (78.9°N, 11.9°E) Archer *et al*. [82], (h) summer in the Baltic Sea off Finland (59.8°N, 23.2°E) Webb *et al*. [88], and (*i*) late-summer in the subtropical North Atlantic (27.9°N, 15.4°W) Archer *et al*. [83]. In order to compare results between experiments, the percentage changes in DMS concentrations between the $pCO_2$ treatments (approx. 350 : 750 µatm, shown as a percentage change on each panel) were calculated using time-integrated DMS concentrations over the duration of each experiment. See electronic supplementary material, table S2. For experiments A, B, C and D, the % response in DMS was calculated from two $pCO_2$ treatments (duplicate mesocosm for (*a*) and triplicate for (*b*–*d*)); for the remaining experiments, the % response was obtained from the linear fit between $pCO_2$ and DMS concentration ($n = 8$, $pCO_2$ treatments for *e*, *f* and *h*; $n = 6$ for *g*). (* note in (*b*) value not significant at 95% confidence interval [87]). (Online version in colour.)

increase in DMS in relation to increased $CO_2$ was reported by Kim *et al*. [61] for their experiment in Korean waters—strongly in contrast to all other studies, and which was attributed to an OA-induced enhancement of dinoflagellate grazers. Such differences highlight the complex responses that can play out within the bounds of a mesocosm experiment, making a general, broad-brush interpretation of results challenging.

The effects of both increased temperature and $CO_2$ on natural communities have been investigated in three experiments using similar sized mesocosms (2.4–2.6 m$^3$), either *in situ* [61,86] or located in controlled-environment containers [114]. In two cases, a reduction in time-integrated DMS with increased $CO_2$ was observed but the elevated temperature had contrasting effects: in the warmer Korean coastal waters (16–19°C), a 2°C elevation in temperature appeared to

decrease net DMS production slightly. Conversely, in the cooler Gulf of St Lawrence, enhanced seawater temperature (10 versus 15°C) led to an increased rate of growth of phytoplankton and bacteria, resulting in elevated DMSP and DMS concentrations. By contrast, Kim *et al.* [61] saw large grazing-induced increases in DMS under both high $CO_2$ and high $CO_2$ + 3°C, with the greatest increase in the high $CO_2$ treatment. These contrasting results could reflect differences in the level of adaptation of the respective communities to natural temperature and $CO_2$ variability, but also highlights the challenges involved in disentangling the complex processes that result in net DMS production.

Finally, it should be recognized that clear discrepancies have arisen in the DMS response to OA between different experimental techniques, which may make interpretation of the overall response challenging. For example, the results of shipboard microcosms contrast strongly with those from mesocosms. However, interpretation of the data can be facilitated by an understanding of the strengths and weaknesses of each technique, and the specific hypotheses each technique is designed to address. Each approach provides valuable information on how OA may influence DMS production in the future. Microcosm experiments are necessarily short term (less than 10 days), so the response to OA is considered to reflect the physiological plasticity of the community, i.e. how well they are adapted to rapidly changing carbonate chemistry, but may not fully capture the effects of shifts in community composition. By contrast, mesocosm experiments are generally much longer (typically approx. 30 days), allowing multigenerational OA-induced changes in taxonomy and community structure to affect DMS concentrations. The microcosm approach may aid in the identification of OA-sensitive regions in terms of DMS production. On the other hand, mesocosm experiments provide some information on how community composition shifts in response to OA may affect the processes controlling the cycle of DMSP and DMS and hence determine their concentrations.

## (b) Nitrogen species

### (i) Ammonia, methylated amines, alkyl nitrates

Oceanic emissions of the soluble trace gas ammonia ($NH_3$) play a role in marine aerosol formation, and the related ammonium ion ($NH_4^+$) provides an inorganic nutrient fundamental to phytoplankton productivity in the surface ocean [104,115]. Both $NH_3$ and its organic analogues, the methylated amines ($R_nNH_{(3-n)}$), are directly affected by changing pH due to their capacity to accept protons (i.e. as bases). A decrease in seawater pH will result in a shift in $NH_3 : NH_4^+$ equilibrium towards $NH_4^+$, and the projected decline in ocean pH from 8.1 to 7.8 by the year 2100 is estimated to reduce the $NH_3$ concentration by 50% [91], decreasing the availability of gas-phase $NH_3$ for ocean–atmosphere gas transfer (table 2). As the recycling of $NH_3$ between the sea and atmosphere is considered to be a major component of the cycling of nitrogen in the marine atmosphere [116], OA has the potential to have a major impact on marine aerosol chemistry over the open ocean, including feedbacks on atmospheric acidity and iron solubilization (e.g. [117,118]) and on particle formation (e.g. [119]).

Recent studies have suggested that marine nitrification of ammonium to nitrate may be significantly inhibited under OA (e.g. [120]) in line with a shift in $NH_3 : NH_4^+$ speciation towards $NH_4^+$. Nitrogen fixation, an important source of new nitrogen to ocean ecosystems will potentially be enhanced under high $CO_2$ conditions ([121] and references therein). A recent meta-analysis of OA studies suggests a decrease of 29% in nitrification, and corresponding increase in nitrogen fixation of 29%, by 2100 under the 'business as usual' emissions scenario [45]. In addition, there is some evidence that $NH_4^+$ uptake by diatoms may be suppressed by OA [122].

Given the complex controls on $NH_4^+$ concentration in the marine environment, it is currently uncertain whether OA will lead to higher $NH_4^+$ concentrations and thus lower ammonia emissions. However, it should be considered whether the simple chemistry that results in a pH-induced shift in $NH_3 : NH_4^+$ equilibrium could on its own alter seawater $NH_3$ concentrations

enough to influence the sea–air exchange of $NH_3$ and amines. Further studies considering the direct effects of OA on the production or consumption of $NH_3$, amines or the atmospherically important alkyl nitrates are required.

### (ii) Terpenes

The marine terpenes (isoprene $C_5H_8$ and monoterpenes $-C_{10}$) occur ubiquitously in the marine environment and have the potential to significantly influence climate via the production of secondary organic aerosol (SOA) [123–125]. There is some recent evidence that OA may induce changes in terpene production by macroalgae, although the direction of response is uncertain and may vary between species, and so requires further investigation [126].

## 7. Atmospheric role: oxidative capacity

### (a) Halocarbons

The surface ocean is a key source of short-lived brominated and iodinated organic compounds (halocarbons) to the atmosphere. Marine emissions of halocarbons, dominated by bromoform ($CHBr_3$), dibromomethane ($CH_2Br_2$) and methyl iodide ($CH_3I$) [127], originate from a range of biological and photochemical processes. These include direct biosynthesis by bacteria (e.g. [128]), phytoplankton (e.g. [70]) and macroalgae (e.g. [13]), and indirect production via reactions between DOM and light [129,130] and/or ozone [23]. Upon entering the atmosphere, halocarbons are rapidly oxidized, yielding short atmospheric lifetimes of less than half a year [131,132], and releasing highly reactive halogen radicals (e.g. I, IO, Br, BrO). These radicals exert an important control on tropospheric ozone [10,133–135], and contribute to the production of new particles and CCN with the potential to influence climate [136].

As marine production of halocarbons is governed by biological processes and the availability of biological substrates (table 2), OA is expected to impact upon their production, with potential feedbacks on atmospheric and climatic processes [137]. However, mesocosm studies have found no obvious effect of high $pCO_2$ and OA on the emission of $CHBr_3$ or $CH_2Br_2$ (e.g. Norwegian fjord [84]), Arctic (Spitsbergen) [85], and brackish waters (Baltic Sea, [88]). By contrast, concentrations of $CH_3I$ were significantly reduced (by up to 67%) under high $pCO_2$ conditions during and after a phytoplankton bloom in temperate waters [84], while in the Baltic Sea, no response was observed [88] (see the electronic supplementary material, table S3 for a summary). Given that these limited studies report conflicting relationships between OA and halocarbon production by surface ocean communities, this is an area that requires further investigation.

A single study [138] has also considered the effects of OA on halocarbon production by tropical seaweeds. Seaweeds are important localized sources of halocarbons [13,14]. In the tropics, biogenic halocarbons contribute disproportionately to stratospheric halogen concentrations and ozone cycling via deep tropical atmospheric convection [139]. Furthermore, seaweed farming is a growing industry in the tropics, so the importance of halocarbon emissions to the atmosphere may increase in the future [138]. Mithoo-Singh *et al*. [138] assessed the response of halocarbon production by five tropical seaweed species to four OA treatments (pH 7.8, 7.6, 7.4, 7.2) relative to ambient (pH 8.0). In general, lower pH resulted in higher halocarbon emission rates, with the effect greatest at the lowest pH treatments (7.4, 7.2). Some resilience within the tested seaweeds to the less severe pH treatments (7.8, 7.6) was apparent, which may result from a degree of adaptation to variation in pH which occurs naturally in coastal waters [140]. However, this should not be taken to represent a linear response given that pH is the $-\log_{10}$ of the $H^+$ concentration. Hence, a greater effect could be expected from the difference between the two lower pH values (7.4, 7.2) than between pH 7.8 and 8.0.

## (b) Oxygenated volatile organic compounds

The small and simple oxygenated VOCs (OVOCs) include methanol, ethanol, propanol, acetaldehyde and acetone. Although predominantly emitted from terrestrial ecosystems [141,142], the oceans play a role as both a source and sink of OVOCs [143–149]. OVOCs affect the oxidative capacity of the troposphere by influencing the ozone budget, consuming hydroxyl (OH) radicals and creating hydrogen oxide ($HO_x$) radicals [150,151]. Although understanding is limited, marine production of OVOCs is linked to biological processes [146,147,152]. For example, acetone and acetaldehyde are thought to be principally produced by photochemical reactions involving the humic component of chromophoric dissolved organic matter (CDOM) [153–156] with up to 68% of gross acetaldehyde production and up to 100% of gross acetone production via this route [147]. Therefore, any OA-induced effect on CDOM characteristics or availability may impact upon the production of these compounds (table 2). Some methanol production may occur via release from living or senescing algal cells [152,157–160], so any OA effects on algal processes could affect the subsequent production of methanol. More research into these gases is required if we are to increase our understanding of the effects of OA on their net production and fluxes.

## 8. Atmospheric role: direct radiative effects

### (a) Nitrous oxide ($N_2O$)

The ocean accounts for approximately one third of natural global emissions of the trace gas $N_2O$ [161,162]. $N_2O$ has the third largest radiative forcing of the anthropogenic greenhouse gases (approx. $300 \times CO_2$ on a molecule per molecule basis) on a global basis [2], and is also a dominant ozone-depleting substance in the stratosphere [163]. It is produced primarily via nitrification in the open ocean, as a by-product of the oxidation of ammonium ($NH_4^+$) to nitrite ($NO_2^-$). $N_2O$ is also produced as a by-product of the reductive denitrification pathway in hypoxic and suboxic environments such as oxygen minimum zones and sediments, where $O_2$ concentrations are sufficient to inhibit $N_2O$ consumption by nitrous oxide reductase enzymes [18,164].

Although there are few studies on the influence of OA on $N_2O$, there is greater insight into its impact on the primary source process of nitrification [165]. Huesemann *et al*. [166] identified a reduction in nitrification rate by up to 90% at pH 6.5, relative to ambient pH, with a linear rate decline across this pH range. Similarly, ammonium oxidation (the first stage of nitrification) decreased to near-complete inhibition at pH 6.5 in experiments using surface waters from the English Channel [167,168]. Although these results are compelling, it should be noted that both studies used lower pH levels than that projected for the next century. Nevertheless, using a more conservative and relevant pH range from 8.09 to 7.42, Beman *et al*. [120] showed unequivocal evidence of an inhibitory effect of OA on nitrification at locations in the Pacific and Atlantic. Conversely, Clark *et al*. [169] found no evidence of a relationship between OA and $N_2O$ or nitrification in near-surface (approx. 5 m) waters in the NW European shelf seas, which they attributed to insignificant production of $N_2O$ in oxic waters. In the only study to investigate the direct impact of OA on $N_2O$, Rees *et al*. [92] recorded a decrease in the $N_2O$ production rate of 2.4–44%, corresponding to a decrease in pH of 0.06–0.4, in cold temperate and polar oceanic waters. This reduction in $N_2O$ yield was directly related to a calculated decrease of 28–67% in $NH_3$ substrate for nitrification that would result from the pH-driven shift in the $NH_3 : NH_4$ equilibrium (see the above discussion). Overall, these results indicate a decrease in $N_2O$ production resulting from the biological response to a physico-chemical transition induced by decreasing pH.

Conversely, Fulweiler *et al*. [170] found that nitrification rates increased with decreasing natural gradients of pH in Narraganset Bay, which they attributed in part to changes in the microbial community in response to competition for $NH_4^+/NH_3$. This is consistent with other suggestions that nitrification may be influenced by OA directly via altered microbial physiology or community composition, or indirectly by changes in the supply of organic material [171]. Hutchins *et al*. [172] speculated that increasing levels of $CO_2$ may lead to an increase in

autotrophic nitrification rates via a $CO_2$ fertilization effect, although this has not been observed in the open ocean. Changes in microbial community composition and abundance in response to OA have been reported [173], particularly for ammonium oxidizing bacteria (AOB) relative to ammonium oxidizing archaea (AOA). Whereas AOB and AOA are ubiquitous and both produce $N_2O$ [174,175], AOA are considered to be the principal nitrifying organisms [176,177], and so an OA-induced shift to AOB may alter marine $N_2O$ production. However, metabolic flexibility may provide some degree of adaptability, with continued growth by coastal AOA reported at a pH of less than 6 [177].

At a lower pH the $N_2O:N_2$ yield of denitrification increases in other environments [178], yet the limited studies in marine systems to date suggest no overall significant effect of OA on denitrification [45]. However, as nitrification carbon and denitrification are coupled in coastal sediments, an OA-induced reduction in nitrification rate may reduce nitrate availability for denitrification leading to a net decrease in $N_2O$ production by both processes [45], although this has yet to be confirmed.

The limited evidence to date suggests that nitrification and associated $N_2O$ production may decrease in the future in response to OA with potential implications for the global marine $N_2O$ source. In a meta-analysis, Wannicke *et al.* [45] concluded that OA might reduce nitrification by $29 \pm 10\%$, consistent with the observed reduction of 3–44% reported by Beman *et al*. [120]. This equates to a decrease in global $N_2O$ production for the next 2–3 decades of 0.06–0.83 Tg N yr$^{-1}$, which is comparable with current global $N_2O$ production from fossil fuel combustion and industrial processes (0.7 Tg N yr$^{-1}$). On the assumption that 50% of the global ocean $N_2O$ source is produced by nitrification [162], Rees *et al*. [92] projected comparable, albeit slightly lower, reductions in oceanic $N_2O$ production. Consequently, the evidence to date suggests the influence of OA may have a small negative feedback on climate change via a reduction in radiative forcing attributed to marine $N_2O$ emissions.

## (b) Methane

Methane ($CH_4$) is a long-lived atmospheric trace gas, which acts as a potent greenhouse gas in the troposphere with a radiative forcing effect, on a molecule per molecule basis, of approximately $25 \times CO_2$ [179,180]. The ocean plays a minor role in the present-day global $CH_4$ budget of the atmosphere [181], contributing a maximum of 10% of the global $CH_4$ burden [182]. Marine $CH_4$ sources are, however, not well constrained, owing to a paucity of observations [183]. Coastal environments including estuaries could account for approximately 75% of the marine source [184], and coastal upwelling areas are also strong sources [185]. Despite the uncertainty regarding the source of $CH_4$ in the surface open ocean [186], there is a potential for direct impacts of OA on $CH_4$ production, via two recently identified methane production pathways, involving DMSP [187] and methane phosphonate [188,189]. $CH_4$ production and consumption mechanisms could also be indirectly impacted by OA, for example, via OA-induced changes in transparent exopolymer particles (TEP) and particle formation [60] that influence methanogenesis in anoxic microsites [190]. $CH_4$ production did not show any OA effects in two studies, but both are currently unpublished: one used Arctic microcosms (A Rees 2019, personal communication) and another used coastal mesocosms (F Deans and C Law 2019, personal communication).

## (c) Carbon monoxide and carbonyl sulfide

The surface ocean is a net source of carbon monoxide (CO), produced via both microbial and abiotic processes, and removed by microbes, mixing and gas exchange [167,168,191,192]. In the atmosphere, CO is a greenhouse gas with a radiative forcing effect of $\sim 2\times CO_2$ (on a molecule for molecule basis). Furthermore, CO indirectly affects the climate by out-competing $CH_4$ in reactions with tropospheric OH radicals, resulting in enhanced concentrations of this far more potent greenhouse gas ($CH_4 \sim 25\times CO_2$, see *Methane*) [193]. Although there are still large uncertainties over the size of the oceanic source of CO, it is likely to be controlled by the quality and quantity

of available CDOM [167,168]. Thus, OA effects on bacterial or phytoplankton processes (that determine the CDOM pool) may alter CO production.

Carbonyl sulfide (OCS) is the most abundant sulfur-containing trace gas in the atmosphere, with marine emissions contributing significantly to the total global budget [194]. OCS is produced in the surface ocean via reactions between UV radiation and CDOM [194,195]. It enters the atmosphere directly via oceanic emission, and indirectly via the oxidation of DMS and carbon disulfide ($CS_2$) in the atmosphere [196–198]. OCS is both a climate-warming greenhouse gas and a climate-cooling aerosol precursor, with the two opposing radiative effects currently in near-balance [199]. However, future changes in the magnitude of sources and sinks may upset that balance [200]. For example, OA may change the oceanic source of OCS via either altering direct emissions from the ocean and wetlands or indirectly via changes in emissions of its precursor gases DMS and $CS_2$.

## 9. Cold and naturally carbonated: trace gas emissions from ocean acidification-sensitive regions

### (a) Polar oceans

Although OA is a global phenomenon, it is progressing with the greatest speed in regions of the ocean that have naturally high dissolved inorganic carbon (DIC) levels and low alkalinity such as high latitude waters of the Southern Ocean and Arctic [28]. In the Arctic Ocean, OA is also accompanied by sea-ice melt water, glacial runoff and river discharge, as well as enhanced terrestrial organic carbon loading, thawing permafrost, gas hydrate destabilization and anthropogenic pollution, which might all further accelerate OA [201–204]. The surface waters of the Arctic Ocean could see a 185% increase in hydrogen ion concentration ($\Delta pH = -0.45$) and basin-wide undersaturation in aragonite ($\Omega_{arag} < 1$) by the end of this century [201,205,206], although with high regional heterogeneity. Based on simulations using the RCP8.5 scenario with the highest concentrations of atmospheric $CO_2$, Popova et al. [112] suggest that the central Arctic, Canadian Arctic Archipelago and Baffin Bay present the greatest rates of acidification and carbonate saturation decline as a result of melting sea ice. By contrast, areas affected by Atlantic inflow including the Greenland Sea and outer shelves of the Barents, Kara and Laptev seas, see minimal decreases in pH and carbonate saturation because diminishing ice cover leads to greater vertical mixing and primary production. OA in the Southern Ocean is primarily driven by the oceanic uptake of anthropogenic $CO_2$ in combination with the naturally strong winter upwelling of DIC-rich, low-alkalinity subsurface waters [207]. Regions of the Southern Ocean already experience sporadic short-term aragonite undersaturation events, the spatial extent and duration of which are expected to accelerate within the next 15–20 years under high $CO_2$ emission scenarios (RCP 8.5) [208].

The polar regions are important for the production of aerosol precursors, such as DMS, that influence CCN production and radiative forcing. In the summertime Arctic atmosphere, marine DMS-derived aerosols significantly contribute to new particle formation events that may influence cloud processes and Arctic atmospheric albedo [7,8,209,210]. The Southern Ocean is a globally important DMS source—regions north of the sub-Antarctic front contribute approximately 15% to global DMS emissions [211], making a significant contribution to DMS-driven secondary aerosol formation [6], and contributing 6–10 W m$^{-2}$ to reflected short wavelength radiation—comparable with the forcing by anthropogenic aerosols in the Northern Hemisphere [212]. Thus, any climate change-induced modification to DMS emissions from polar regions could influence radiative forcing at both regional and global scales. The modelling studies by Six et al. [43] and Schwinger et al. [44] indeed show a significant radiative forcing and surface temperature increase due to OA-induced reductions in polar DMS production under high emissions pathways (e.g. a 0.86 W m$^{-2}$ reduction in reflected short wave radiation south of 40°S in the study of Schwinger et al.).

A small number of experimental studies report the effects of OA on DMS in polar waters. In a mesocosm experiment in Kongsfjorden, Svalbard Archipelago (78°N), a 35% decrease in DMS at 750 µatm was attributed to a decrease in bacterial DMSP-to-DMS yields [82]. Similarly, in a microcosm experiment conducted in Baffin Bay, Canadian Archipelago (71°N), a 25% decrease in DMS at approximately 1500 µatm was found, attributed to an OA-related increase in sulfur demand by the bacterial assemblage [81]. These limited results suggest that the net production of DMS during the productive summer season in the Arctic could decrease via bacterioplankton-mediated processes with ongoing OA. In contrast to these previous experiments, in a series of shipboard microcosm experiments in the Arctic and Southern Ocean surface waters, little biological effect and minimal DMS response to OA was observed [90], suggesting a high level of resilience to a changing carbonate chemistry environment within the sampled communities. This agrees with previous evidence that polar microbial communities may be adapted to a changing carbonate chemistry environment as they experience strong natural fluctuations in pH (over the range 7.5–8.3) over diurnal/seasonal and local/regional scales [112,213–215]. However, it cannot be excluded that variations in community responses may be linked to differences in experimental approaches used, as previously described (see section on 'Reconciling differences within and between experimental techniques').

## (b) Eastern boundary upwelling systems

Eastern Boundary Upwelling Systems (EBUS) are considered particularly susceptible to OA, given the combined effects of their naturally high DIC concentrations and enhanced uptake of anthropogenic $CO_2$ [216]. Characteristic examples of EBUS include the California and Peru/Humboldt EBUS in the Pacific, and the Canary and Benguela EBUS in the Atlantic. Cold DIC-rich subsurface waters are upwelled to the surface layer by trade wind forcing at seasonal and interannual timescales, lowering the pH of surface waters relative to open ocean surface waters [216]. This enhances the rates of OA within such systems relative to the global surface ocean. Recent data from the California Upwelling System, using a proxy record of fossil foraminifera calcification response, have shown a 35% decrease in $[CO_3{}^{2-}]$ and a drop in pH of 0.21 units since pre-industrial times, which exceeds the global mean decline by a factor of two [217]. Due to the high decomposition rate of organic matter as well as the input of equatorial low-$O_2$ water masses, oxygen minimum zones (OMZs) exist in these coastal areas affecting the regional oxygen, nitrogen, carbon and sulfur cycles through nitrification, denitrification, anammox and sulfate reduction processes and influencing local trace gas production. EBUS are considered to be 'hot spots' for emissions of greenhouse gases ($N_2O$, $CH_4$) and reactive species such as DMS and $H_2S$ [18,218–224].

The addition of anthropogenic $CO_2$ into the already corrosive waters of EBUS could rapidly push these systems closer to critical thresholds, such as aragonite undersaturation [216]. Indeed, pH values as low as 7.6 and 7.7 have been measured in the Californian upwelling, accompanied by a shoaling of the aragonite saturation horizon by about 50 m since preindustrial times [216]. This leads to periodic upwelling of corrosive waters during the summer months [216,225,226], with the potential to impact upon ecologically and economically important species [216,225,227,228]. Despite the observational work that has been undertaken, little experimental work has been conducted on the implications of OA in EBUs (e.g. [229]). Thus, the OA impact on trace gas production in these regions is still highly uncertain, but potentially in line with the responses observed in other regions, as described above.

## 10. Ocean acidification, warming and deoxygenation: the multi-stressor effects on marine trace gases

OA is not occurring in isolation to other global environmental changes. In addition to having taken up approximately 28% of the excess anthropogenic $CO_2$ since 1750, the ocean has also

absorbed approximately 93% of the excess heat over the past 45 years [30]. Both processes profoundly modify the physical and chemical environment experienced by marine organisms. Warming enhances biological rates [230,231] and decreases the solubility of gases, resulting in decreasing global ocean oxygen inventories [29]. Warming and freshening enhances surface ocean stratification [232], which in turn decreases mixed layer depth and reduces the entrainment of nutrients into the euphotic layer, while resulting in higher levels of irradiance experienced by organisms [29]. This alleviates light limitation at high latitudes, but enhances nutrient limitation at low- to mid-latitudes. Reductions in nutrient entrainment may be compensated for by the atmospheric deposition of anthropogenic aerosols, which itself could be countered by future improvements in air quality standards [233]. Thus, $CO_2$-driven changes in seawater carbonate chemistry occur simultaneously with warming, deoxygenation, localized freshening of the ocean and changes to nutrient dynamics [234,235]. Numerous modelling studies have addressed future changes in marine ecosystems and biogeochemistry in response to these drivers, either in isolation or combined [29,49,236], yet very few have focused on trace gas emissions and OA-related feedbacks to the Earth system [42–44].

## (a) Changing dimethyl sulfide emissions in response to ocean acidification: earth system feedbacks

Although experimental data provide useful information on the potential future DMS response to OA, these data become most powerful when included in an Earth System Model (electronic supplementary material) to facilitate upscaling and estimation of feedbacks of projected changes in DMS emissions on future climate (figure 3). So far, two studies have used electronic supplementary material to provide evidence for a potential positive climate feedback arising from pH-sensitivity of DMS production [43,44]. At the end of the current century, the electronic supplementary material showed major pH-induced reductions in DMS production for areas of high biological production, such as the upwelling equatorial Pacific and other EBUS, the eddy-driven upwelling in the Southern Ocean around 40°S and the subpolar biome in the North Atlantic [43,44]. Both studies revealed a subsequent significant radiative forcing and surface warming in response to the decreased DMS flux to the atmosphere and subsequent changes in aerosol and cloud properties. Schwinger et al. [44] used a fully interactive model, able to simulate a range of feedbacks: they found a global linear relationship between pH-induced changes of DMS sea–air fluxes and a transient surface temperature change of $-0.041°C\,TgS^{-1}\,yr^{-1}$, driven by reductions in global DMS emissions. These model experiments were conducted with a high-emission scenario (RCP8.5) as a baseline, leading to average surface pH reductions of 0.44 and 0.73 units in 2100 and 2200, respectively. The corresponding reduction of DMS fluxes (assuming the 'medium' pH-sensitivity of DMS production of [43]) is $4\,Tg\,S\,yr^{-1}$ (17%) in 2100 and $7.3\,Tg\,S\,yr^{-1}$ (31%) in 2200. The simulated additional surface warming has a north–south gradient with much stronger surface warming in the Southern Hemisphere due to the larger area covered by ocean (figure 3).

Both models described here are parametrized using the empirical relationship between pH and DMS observed in a number of mesocosm studies [77,82,84,87], while recognizing that the level of understanding of the DMS response to OA within these experiments is limited. It should also be noted that these data consider OA as a single stressor, with a complete lack of information for other key climate stressors such as ocean warming. Furthermore, our interpretation of the DMS response between mesocosm studies is confounded by inconsistencies in composition and physiological status of starting communities and experimental set-up (e.g. volume of seawater, method of acidification, inorganic nutrient additions, inclusion/exclusion of higher trophic levels, light and UV cycles, mixing, wall effects/cleaning) that make it difficult to draw direct comparisons (see discussion below). To increase the accuracy of model outcomes and facilitate a better understanding of the future feedbacks and climatic effects, improved comparison and integration of all DMS data from mesocosm experiments are required. For example, normalizing

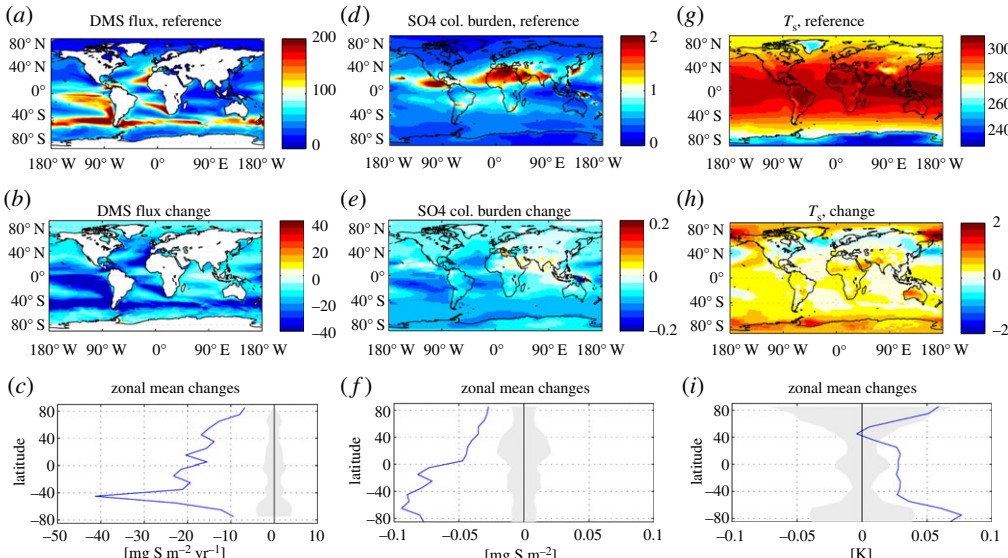

**Figure 3.** Outputs from the fully interactive Earth system model run from Schwinger *et al.* [44]. The top row of panels show DMS sea–air flux (*a*), sulfate aerosol burden (*d*) and surface temperature (*g*) in the reference simulation (no sensitivity of DMS fluxes to OA). The corresponding changes in a simulation assuming a decrease of DMS production with increasing pH are shown in the middle row of panels (*b,e,h*) and zonal mean changes are depicted in the bottom row (*c,f,i*). The grey shaded area in the zonal mean plots gives the range of natural variability (defined as the standard deviation of the zonal mean found in the control run). (Online version in colour.)

for differences in experimental design, community structure and carbonate chemistry dynamics could lead towards a more accurate empirical relationship between pH and DMS. Where the DMS representation in a model is detailed enough, e.g. [237], it would be beneficial to include the effects of OA on the processes controlling DMS production in the surface ocean, for example, using data from short term, small-scale experiments, such as shipboard microcosms (e.g. [80,81,90]).

Finally, there is a large gap in our understanding of the response of net DMS production to other climate stressors in community-level experiments, in particular the response to increasing temperatures. Three mesocosm experiments to date have considered the combined effects of OA and temperature [61,86,114]. Our understanding of the multi-stressor response of DMS, and other trace gases, would be improved with a greater understanding of such processes. For example, Dani & Loreto [22] hypothesize that phytoplankton isoprene emitters favour warmer latitudes, as opposed to cold water-favouring DMS emitters. This may imply that as warmer waters extend towards higher latitudes with climate change, there could be an increase in geographical extent of oceanic isoprene producers to the detriment of DMS producers. However, further work is now required to fill such gaps in our understanding.

## (b) Marine nitrogen cycle Earth system feedbacks

Changes to the future ocean source of $N_2O$ have been evaluated as a direct consequence of global warming-driven changes in ocean circulation and productivity [238] and combined with anthropogenic nitrogen deposition [239]. Both studies report a decrease in $N_2O$ emissions by 2100 of between 4–12% [238] and 24% [239]. The decrease results from both a net reduction in $N_2O$ production and an increase in $N_2O$ storage driven by enhanced stratification and reduced $N_2O$ sea–air flux. The reduction in net $N_2O$ production in both studies is largely driven by reduced primary production and export production resulting in decreased water column nitrification, changes in ocean circulation in response to global warming and atmospheric $N_2O$

concentrations, combined with expansions of OMZs and associated increases in water column denitrification [238,239]. On millennial timescales under sustained anthropogenic climate forcing, Battaglia & Joos [240] project increases in $N_2O$ production of 21% due to deoxygenation and elevated remineralization fluxes. Under steady-state conditions, these millennial increases in $N_2O$ emissions are shown to cause a small positive climate feedback ($0.004\,W\,m^{-1}\,K^{-1}$ for RCP8.5, [240]).

However, given the limited evidence for direct effects of OA on $N_2O$ production, the model studies assessing the future evolution of $N_2O$ emissions have not, so far, included the effects of OA. Similarly, effects of deoxygenation on $N_2O$ production in a warming ocean remain underexplored, largely due to persistent biases in the climatological representation of OMZs and in reproducing the expansion of low oxygen waters in electronic supplementary material [241,242]. Modelling studies that apply parametrizations based on mesocosm studies to describe the effect of OA on the stoichiometry of organic matter [243] have shown that OA can exacerbate ocean deoxygenation via enhanced C : N ratios in organic matter [244,245]. The higher C : N ratio in organic matter would constitute a negative feedback on atmospheric $CO_2$ through the strengthening of the biological pump. However, the enhanced $O_2$ utilization during remineralization would promote the production of $N_2O$, a positive feedback to the Earth radiative budget, which would offset the first one. The change in the stoichiometry of organic matter in response to OA remains, however, to be confirmed by further studies [246,247]. In summary, current model projections suggest a future decline in global marine $N_2O$ emissions, and small negative feedback to climate change. However, these analyses do not account for the influences of OA and have limited capability to assess key influences on the marine nitrogen cycle such as deoxygenation. Elucidating these influences will require a combination of improved process knowledge and incorporation of this into more representative biogeochemical process models.

## (c) Key areas of future research on multi-stressors

While models are vital to exploring responses of trace gas emissions to multiple stressors, the development of adequate parameterization depends on experimentally evidenced process understanding. Figure 4 summarizes our knowledge of the anticipated direct and indirect effects of multiple stressors on trace gas production. These stressors operate at global scales, including warming and acidification, and at regional scales, such as in coastal waters and polar regions. Figure 4 also indicates the inferred trace gas response (increased or decreased production) to each stressor, although many of these are based on limited observations. Whereas some stressors, such as eutrophication, are considered to have a primarily stimulatory effect on trace gas production, others can have both positive and negative impacts. For example, warming may stimulate trace gas production by enhancing metabolic rates and reducing oxygen availability, but may also reduce phytoplankton diversity potentially reducing production of taxa-specific trace gases such as DMS and halocarbons. From the perspective of individual trace gases, the production of some, such as methane, may increase in response to most stressors, whereas the majority of trace gases may show increased or decreased production depending on the stressor. Perturbation experiments on marine ecosystems that assess multiple stressors are still rare, and consequently, there is little information as to how they influence trace gas production. Although the overriding trend in marine multiple stressor studies is synergistic, relative to rates of the individual stressors [248], multiple stressors with opposing impacts may cancel each other out, or alternatively one may dominate. It is recommended that future studies of trace gas production consider the impact of multiple stressors [249] using region-specific projections for climate variables (see [250]).

## 11. Conclusion

The potential for marine trace gas emissions to influence and impact atmospheric chemistry and climate are substantial. The changes in net production of some trace gases such as DMS and $N_2O$, indicated in OA studies and models, point to potentially large and globally significant

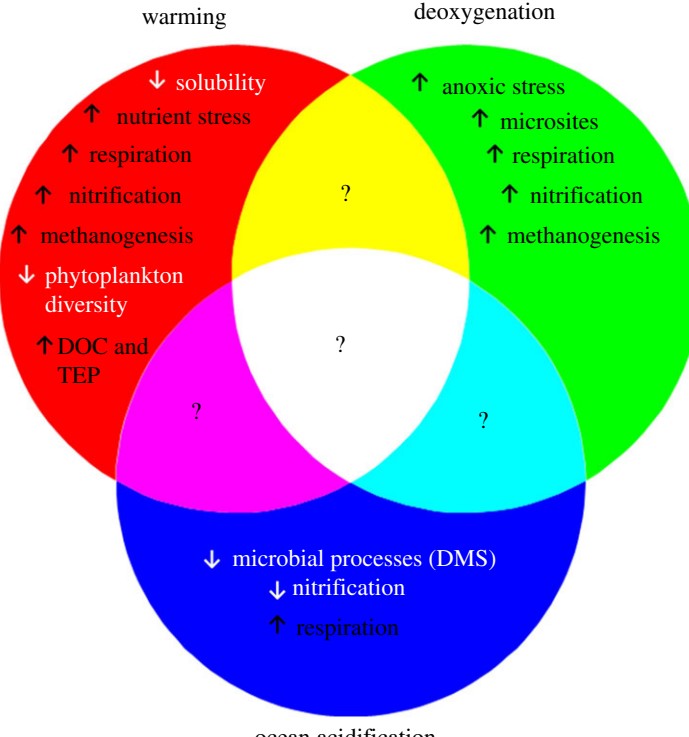

| Stressor | CO₂ | N₂O | CH₄ | DMS | NH₃ | Other trace gases (halocarbons, CO, OCS, CS₂, OVOCs, isoprene) | Indirect factors affecting trace gas production & emission |
|---|---|---|---|---|---|---|---|
| *Directly influenced by climate* | | | | | | | |
| **Warming** | ↓ : ↓solubility<br>↑ : ↓ primary production<br>↑ : ↑ respiration<br>↑ : ↑ DOM & TEP | ↓ : ↓ solubility<br>↑ : ↑nitrification | ↓ : ↓ solubility<br>↑ : ↑methanogenesis<br>↑ : ↑ hydrate decomposition | ↓ : ↓ solubility<br>↑ : ↑ nutrient stress<br>↓ : phytoplankton population shifts<br>↑/↓ : changes to bacterial processes | ↓ : ↓ solubility | ↓ : ↓ solubility (all)<br>↑ : global phytoplankton distribution shifts (isoprene) | ↓ : ↑surface stratification & decreased nutrient supply to surface ocean<br>↓ : ↓ C fixation & export<br>↓ : changes to ventilation & thermohaline circulation<br>↑ : ↑ winds & storm intensity<br>↑ : ↑ dust deposition |
| **Ocean Acidification** | ↑ : ↑ DOM & TEP<br>↔ : ↔ primary production | ↓ : ↓ nitrification | ↔ : no response | : phytoplankton population shifts<br>↓ : ↓ phytoplankton biomass<br>↑ : ↑ physiological stress<br>↑/↓ : ↑/↓ bacterial processes<br>↑ : ↑ grazing on phytoplankton | ↓ : ↓ NH₄⁺ (phase shift) | : phytoplankton population shifts (iodocarbons)<br>↔ : no response (bromocarbons)<br>? : CO, OCS, CS2, isoprene | ↑↓ : ↑↓ primary production & C export<br>↓ : ↓ export production (decreased carbonate production)<br>↑ : ↑ trace metal availability (phase shifts) |
| **Hypoxia** | ↑ : ↑respiration | ↑ : ↑nitrification<br>↑↓ : ↑↓ denitrification | ↑ : ↑ anaerobic microsites<br>↑ : ↑ methanogenesis | ↑ : ↑ anoxic stress | ↑ : ↑ammonification | ? | ↓ : ↓ nutrient supply in upwelling regions |
| *Locally variable/indirectly influenced by climate* | | | | | | | |
| **Increased UV-R** | ↓ : ↓ respiration<br>↑ : ↑ photoproduction | ↓ : ↓nitrification | ↔ | ↑ : ↑ photooxidative stress<br>↓ : ↓ phytoplankton biomass | ? | ↑ : ↑ photoproduction (CO, halocarbons, isoprene)<br>↑ : ↑ photooxidative stress<br>↓ : ↓ phytoplankton biomass | ↓ : ↓ phytoplankton biomass |
| **Increased nutrient loading** | ↑ : ↑respiration<br>↑ : ↑ DOC | ↑ : ↑ nitrification<br>↑ : ↑denitrification | ↑ : ↑ anaerobic microsites<br>↑ : ↑ methanogenesis | ↑ : ↑ primary production<br>↑ : ↑ phytoplankton biomass | ↑ : ↑ DON | ↑ : ↑photoproduction (CO, halocarbons, isoprene) | ↑ : ↑ primary production & C export<br>↑ : ↑ occurrence of HABs<br>↑ : ↑hypoxia |
| **Increased particle loading** | ↑ : ↑respiration<br>↑ : ↑ DOC | ↑ : ↑nitrification<br>↑ : ↑denitrification | ↑ : ↑ anaerobic microsites<br>↑ : ↑ methanogenesis | ↓ : ↓primary production | ↑ : ↑ DON | ↓ : decreased photoproduction (CO, halocarbons, isoprene) | ↓ : ↓ bioturbation<br>↑ : ↑hypoxia |

**Figure 4.** Summary of our knowledge on multiple stressors and their anticipated direct and indirect effects on trace gas production. Coloured arrows represent known/anticipated trace gas response (red, increase; blue, decrease; green, no net change), and black arrows describe the direction of change of the related process. HABs, harmful algal blooms; TEP, transparent exopolymer particles; DON, dissolved organic nitrogen. (Online version in colour.)

modifications to sea–air fluxes. This could lead to either warming (e.g. lower DMS emissions) or cooling (e.g. lower $N_2O$ emissions) effects on climate. Where data for other trace gases are scant, we cannot yet be confident in the direction of change, but we can have greater certainty that there is the potential for impacts on net production, and so chemistry and climate, with global-scale effects.

However, relative to other aspects of marine biological and ecological research, the field of marine trace gas production is under studied. Even our understanding of the basics could be improved, such as the processes driving production and cycling within the surface ocean. These knowledge gaps can limit our ability to design appropriate experiments or to interpret findings in the context of OA. Furthermore, even where some data are available, the limited mechanistic representation of biological and biogeochemical processes in electronic supplementary material limits the predictive capability of future trace gas production and emissions, and related climate effects. Inconsistencies in the effect of OA on trace gas production result from the complexity of trace gas cycling, with the involvement of multiple production and loss processes (e.g. phytoplankton species composition, bacterial processes and grazing activities). Further complications arise when the potential for both direct and indirect effects on trace gas production is considered, and as with other aspects of OA research, the indirect effects are more challenging to pin down (figure 4). Finally, interpretation of experimental data and projections in terms of atmospheric chemistry and climate are complicated by trace gas sensitivity to other climate change stressors (warming, deoxygenation and eutrophication), some of which may be more important determinants of production and emissions.

Of course, understanding the biological mechanisms (and their regulation) will be crucial for interpreting the trace gases response to OA, using both model organisms in the laboratory and natural communities within field experiments, and addressing the current shortcomings requires an improved experimental approach. Although short-term OA studies provide useful information on the physiological plasticity of surface ocean communities and associated trace gas production, and existing levels of adaptation to fluctuations in carbonate chemistry, such experiments cannot accommodate the potential for evolutionary adaptation of planktonic communities (e.g. [251–254]). Therefore, it would be beneficial to carry out longer term experimental studies, encompassing multiple generations, in order to detect adaptation of planktonic communities to OA and other climate change stressors. Such adaptation in phytoplankton becomes evident after only a few hundred generations, representative of a period of approximately 6–12 months (e.g. [251]). Parallel measurements of process rates and standing stocks of trace gases would provide greater insight into the role of OA in influencing trace gas production. However, the implementation of long-term experiments of this kind are likely to be limited to culture conditions using isolated strains, and thus at the expense of other important ecological and biogeochemical interactions (see [52]). Ecological level experiments will still involve a trade-off in terms of duration and number of generations, but will continue to provide important information on the role of species interactions and succession on trace gas production. Both experimental approaches could integrate multiple stressors, thus closing some of the gaps in our understanding of the trace gas response to climate change. An enriched experimental understanding could be complemented by improved surface ocean measurements. To this end, we recommend that future surface ocean trace gas measurements are accompanied by quantification of at least two components of the carbonate system, so that global databases can be used to relate spatial variability in trace gas concentrations to variations in surface ocean pH. This would greatly increase our understanding of the influence of the carbonate system, including the physical and biogeochemical processes that control pH, on trace gas concentrations in the surface oceans.

To provide reliable projections of future marine emissions of climate-relevant gases, studies will need to characterize and quantify the nature of the adaptation and/or resilience of diverse trace gas-producing communities. The complexities of these investigations are compounded by the multitude of environmental changes, including OA, that affect these communities. Such studies would be complex in design and implementation, and require community-wide

collaborative efforts, involving researchers from multiple disciplines and significant levels of financial investment (see [52,249,255]). However, given success, this would improve our understanding of the longer term effects of OA on the biological and biogeochemical processes involved in trace gas production, and build an improved mechanistic representation of these processes into models. This would be a much-needed improvement on the current use of empirical relationships and could lead to a step change in our predictive capability.

Data accessibility. This article does not contain any additional data.

Authors' contributions. F.E.H., P.S. and M.G. coordinated the development of the review, and F.E.H. led the writing effort. All co-authors participated in the workshop that stimulated the development of this review, and contributed to key discussions. F.E.H., P.S., M.G., S.D.A., C.S.L., K.L., G.M., M.L., O.A., A.P.R., M.J., H.N., F.K., I.D., L.B., N.G., J.S. and E.B. contributed text to the review. P.W., P.S.L., J.C.M., T.J. and R.D. provided feedback and detailed edits on the text.

Competing interests. We declare we have no competing interests.

Funding. All authors thank the International Science Council's Scientific Committee on Oceanic Research (SCOR), the US National Science Foundation, and the Global Atmosphere Watch of the World Meteorological Organization, the International Maritime Organization and the University of East Anglia for their support. F.E.H. was funded via the Natural Environment Research Council (UK Ocean Acidification grant no. NE/H017259/1). P.S. acknowledges funding from the European Union's Horizon 2020 research and innovation programme under grant agreement no. 641816 Coordinated Research in Earth Systems and Climate: Experiments, kNowledge, Dissemination and Outreach (CRESCENDO). Financial support for S.D.A. was provided by the National Science Foundation, United States (NSF Project OCE-1316133). L.B. acknowledges support from the H2020 CRESCENDO grant no. 641816 and the MTES/FRB Acidoscope project. F.K. received funding from the Phase II Higher Institution Centre of Excellence (HICoE) Fund, the Ministry of Education Malaysia (IOES-2014F) and the University of Malaya Top 100 Research University grant no. TU001D-2018. C.S.L. was supported by funding from the New Zealand CARIM (Coastal Acidification: Rate, Impacts and Management) project and NIWA SSIF. K.L. was supported by the Global Research Laboratory Program (2013K1A1A2A02078278) funded by the National Research Foundation of Korea. M.L. was supported by SCOR and Natural Sciences and Engineering Research Council of Canada (NSERC) Discovery Grant Program. A.P.R. was supported by funding from the Natural Environment Research Council (UK Ocean Acidification grant no. NE/H017259/1 and Changing Arctic Ocean grant no. NE/R012830/1). J.S. acknowledges funding from the Research Council of Norway (grant no. 295046). P.W. was supported by the NERC-Defra Shelf Sea Biogeochemistry programme.

Acknowledgements. This paper resulted from the deliberations of United Nations GESAMP Working Group 38, 'The Atmospheric Input of Chemicals to the Ocean', and culminated from a Working Group 38 workshop held at the University of East Anglia, Norwich, UK, on *Impacts of ocean acidification on fluxes of non-CO₂ climate-active species.* We are grateful to GESAMP for making this workshop possible and providing the opportunity to bring together the experts in this field. All participants to the workshop have contributed to the development and writing of this paper and are thus included as co-authors.

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
