## [Reviewer comments · Proceedings. Mathematical, Physical, and Engineering Sciences]

Review History

RSPA-2019-0769.R0 (Original submission)

Review form: Referee 1

Is the manuscript an original and important contribution to its field?

Excellent

Is the paper of sufficient general interest?

Good

Is the overall quality of the paper suitable?

Excellent

Can the paper be shortened without overall detriment to the main message?

Yes

Do you think some of the material would be more appropriate as an electronic appendix?

No

Do you have any ethical concerns with this paper?

No

Recommendation?

Accept with minor revision (please list in comments)

Comments to the Author(s)

This is a comprehensive review of the impacts of ocean acidification on different phytoplankton and macrophytes and their production (and by inference emission) of trace gases important for climate and atmospheric chemistry. This is a young field, and the review shows that few conclusive statements could be made based on the paucity of focused studies, and the diverse designs of the experimental studies. The review concludes with suggestions for remedying the situation.

Major comments

1. To my surprise, the values of pH are barely mentioned in this review about ocean acidification. It would be very helpful to include the pH ranges in the different studies (P6, L18 onwards, and Tables 1 and 2). This would help this reader put the various results in some context.

2. Figure 4 is an important summary figure. The last column "Indirect factors affecting trace gas production" is confusing to this reader. It would be helpful to state, for example, "increased surface stratification and decreased nutrient supply" to get a sense of the indirect factors. It is not clear to me why sea level rise would decrease trace gas production. Also, what are the indirect factors associated with Toxins (last row, last column)?

3. The conclusions of the review seem generic – the system is complicated, we need more studies, longer studies, etc. The manuscript would benefit from a targeted set of recommendations – for example, the pH ranges and durations (what is the lifetime of a phytoplankton/macrophyte generation?) of experiments, the comprehensive suite of observations needed in OA sensitive regions. This reader infers that the target of the experimental studies may be pH of 7.8 predicted for 2100 (P11-L31), as some studies with pH as low as 7.2 (P13-L32) and 6.5 (P15-L10) are deemed beyond the expected range. Yet pH 7.6 and 7.7 have already been measured in the California upwelling system (P20-L8), and natural pH fluctuations over the range of 7.5 to 8.3 have been observed in the Arctic and Southern Oceans (P19-L19).

4. What are the key phytoplankton and macrophyte species for the trace gases under consideration? Do they have different pH thresholds for tolerance/change?

5. The title "Unravelling the impacts..." seems aspirational. The review has not clarified the complicated system for this reader.

Other comments

6. Increases in DIC are not the only cause of acidification. Temperature increases (at constant DIC) also lowers pH.

7. P7-L27 onwards. What about effects on clouds?

8. P17-L16-18. CO is a weak GHG, as stated. However, CO competes with CH₄ for OH, and so high CO in the atm means a longer CH₄ lifetime.

9. P18 L32. 6-10 W/m² of reflected sunlight is the total response (background+perturbation), and not the forcing (perturbation since 1800's). A reader may confuse this with discussion of radiative forcing in the section on aerosols. It would be better to show the forcing from the modeling studies (P19-L3-4).

10. P21-L19: Please give numbers in the text: decreases in pH and DMS flux, so that $-0.041C/TgS$ could be put in context.

11. Additional references:

- a. pH measurements at Hawaii Time Series
- b. US Northeast coast - Rheuban et al. JGR Oceans 2019.
- c. Osborne EB et al. Nature Geoscience Dec 2019 – California current ecosystems

Review form: Referee 2

Is the manuscript an original and important contribution to its field?

Excellent

Is the paper of sufficient general interest?

Excellent

Is the overall quality of the paper suitable?

Good

Can the paper be shortened without overall detriment to the main message?

Yes

Do you think some of the material would be more appropriate as an electronic appendix?

No

Do you have any ethical concerns with this paper?

No

Recommendation?

Accept with minor revision (please list in comments)

Comments to the Author(s)

This manuscript is a review of how ocean acidification influences trace gas cycling and exchange. The authors outline the types of experiments to date and the major findings, covering a variety of trace gases. Their focus, however, is on dimethylsulfide (DMS), as it is the most widely studied biogenic trace gas in relation to ocean acidification. The authors also give recommendations on conducting future studies. Such a review is an important contribution to our understanding of trace gas cycling in a changing environment and useful for the greater community, as many of the findings to date can be confusing to interpret. The authors are clearly experts in the field and have been comprehensive in their treatment of the topic. I recommend that the review be published after the minor comments below are addressed.

Specific comments:

The manuscript repeatedly discusses biogenic trace gas production, but neglects loss processes almost entirely. I think there should be a bit of text addressing how oceanic loss processes are also influenced by ocean acidification.

Page 8, line 10 – Typo, missing word in “...with blooms of this species associated with the release vast quantities of DMS...”

Page 9, lines 9-15 – Why is DMS the most widely studied compound? Maybe it is worth it to say a few words about this?

Pages 9 and 10 – There is no information in the description of DMS/P/O cycling about cell lysis

from death, grazers etc that promotes DMS formation, but on page 10, line 13 there is a reference to grazers. It would be good to include something in the text earlier to give context to this point.

Page 11, line 3 - Typo, too many words in "...hypothesis may not to be universally...".

Page 12-13, section Reconciling differences within and between experimental techniques - Is there a take home message to this section?

Page 16, lines 10-12 - Missing reference to Marandino et al. (2005) and Schlundt et al. (2017)
Schlundt, C., Tegmeier, S., Lennartz, S. T., Bracher, A., Cheah, W., Krüger, K., Quack, B. and Marandino, C. A. (2017) Oxygenated volatile organic carbon in the western Pacific convective centre: ocean cycling, air-sea gas exchange and atmospheric transport *Atmospheric Chemistry and Physics*, 17. pp. 10837-10854. DOI: 10.5194/acp-17-10837-2017.

Marandino, C. A., de Bruyn, W. J., Miller, S. D., Prather, M. J., and Saltzman, E. S. (2005) Oceanic uptake and the global atmospheric acetone budget *Geophysical Research Letters*, 32 (15). DOI 10.1029/2005GL023285.

Page 18, lines 24-25 - Is this negative feedback from the combined effect of greenhouse warming and ozone depletion in stratosphere?

Page 19, section on CO and COS - The role of CDOM in COS ocean cycling is completely ignored and this should be included as CDOM is the main oceanic precursor.

Page 22, lines 2-5 - missing reference to Arevalo et al. (2015)

Arevalo-Martínez, D. L. , Kock, A. , Löscher, C., Schmitz, R. A. und Bange, H. W. (2015) Massive nitrous oxide emissions from the tropical South Pacific Ocean. *Nature Geoscience*, 8 (7). pp. 530-533. DOI 10.1038/ngeo2469.

Page 22, line 15 - missing second parenthesis after citation

Page 22, section Ocean acidification, warming, and deoxygenation: the multi-stressor effects on marine trace gases- Should there be some mention of atmospheric deposition (in the context of reduced mixing)?

Page 27, line 2 - missing second parenthesis after citation

Figure 1 - COS and CS₂ have large net ocean sources that are not included in the schematic (can be included in the DOM area).

Figure 2 - Is there any relationship between DMS (initial) concentrations and percent changes? The amounts in panel f are huge. Is this correct? I did not check the original citation.

Figure 4 - Why is uv-r considered local? In the chemical titles - what exactly is the title of the short lived box; isoprene is one HC; where is COS? In terrestrial environments (also in the ocean), there is a positive correlation of isoprene with temperature, but this is not included here. What is the difference between blank boxes and question marks? Are microbes and phytoplankton used interchangeably (ignoring bacteria)? Because bacterial metabolism may increase with increasing temperature, which may have the effect of increasing DMS, which is not included in the table.

Table 2 - Isoprene and CS₂ are not included anywhere.

Supplementary material

Table S1 - perhaps add information to convey season, location (e.g. temperature)

Table S2 - add season

Table S3 - typo, make CO₂ subscript; maybe a mistake last line, as text says methyl iodide in Bergen experiment, but here states CH₂I₂ in Svalbard.

Decision letter (RSPA-2019-0769.R0)

21-Jan-2020

Dear Dr Hopkins,

On behalf of the Reviews Editor, I am pleased to inform you that your Manuscript RSPA-2019-

0769 entitled "Unravelling the impacts of ocean acidification on marine trace gases and the implications for atmospheric chemistry and climate" has been accepted for publication subject to minor revisions in Proceedings A. Please find the referees' comments below.

The reviewer(s) have recommended publication, but also suggest some minor revisions to your manuscript. Therefore, I invite you to respond to the reviewer(s)' comments and revise your manuscript. It is a condition of publication that you submit the revised version of your manuscript within 7 days. If you do not think you will be able to meet this date please let me know in advance of the due date.

To revise your manuscript, log into <https://mc.manuscriptcentral.com/prsa> and enter your Author Centre, where you will find your manuscript title listed under "Manuscripts with Decisions." Under "Actions," click on "Create a Revision." Your manuscript number has been appended to denote a revision.

You will be unable to make your revisions on the originally submitted version of the manuscript. Instead, revise your manuscript and upload a new version through your Author Centre.

IMPORTANT: Your original files are available to you when you upload your revised manuscript. Please delete any redundant files before completing the submission process.

In addition to addressing all of the reviewers' and editor's comments, your revised manuscript **MUST** contain the following sections before the reference list (for any heading that does not apply to your work, please include a comment to this effect):

- Acknowledgements
- Funding statement

See <https://royalsociety.org/journals/authors/author-guidelines/> for further details.

When uploading your revised files, please make sure that you include the following as we cannot proceed without these:

- 1) A text file of the manuscript (doc, txt, rtf or tex), including the references, tables (including captions) and figure captions. Please remove any tracked changes from the text before submission. PDF files are not an accepted format for the "Main Document".
- 2) A separate electronic file of each figure (tif, eps or print-quality pdf preferred). The format should be produced directly from original creation package, or original software format.
- 3) Electronic Supplementary Material (ESM): all supplementary materials accompanying an accepted article will be treated as in their final form. Note that the Royal Society will not edit or typeset supplementary material and it will be hosted as provided. Please ensure that the supplementary material includes the paper details where possible (authors, article title, journal name). Supplementary files will be published alongside the paper on the journal website and posted on the online figshare repository (<https://figshare.com>). The heading and legend provided for each supplementary file during the submission process will be used to create the figshare page, so please ensure these are accurate and informative so that your files can be found in searches. Files on figshare will be made available approximately one week before the accompanying article so that the supplementary material can be attributed a unique DOI.

Alternatively you may upload a zip folder containing all source files for your manuscript as described above with a PDF as your "Main Document". This should be the full paper as it appears when compiled from the individual files supplied in the zip folder.

Article Funder

Please ensure you fill in the Article Funder question on page 2 to ensure the correct data is collected for FundRef (<http://www.crossref.org/fundref/>).

Media summary

Please ensure you include a short non-technical summary (up to 100 words) of the key findings/importance of your paper. This will be used for to promote your work and marketing purposes (e.g. press releases). The summary should be prepared using the following guidelines:

*Write simple English: this is intended for the general public. Please explain any essential technical terms in a short and simple manner.

*Describe (a) the study (b) its key findings and (c) its implications.

*State why this work is newsworthy, be concise and do not overstate (true 'breakthroughs' are a rarity).

*Ensure that you include valid contact details for the lead author (institutional address, email address, telephone number).

Cover images

We welcome submissions of images for possible use on the cover of Proceedings A. Images should be square in dimension and please ensure that you obtain all relevant copyright permissions before submitting the image to us. If you would like to submit an image for consideration please send your image to proceedingsa@royalsociety.org

Once again, thank you for submitting your manuscript to Proceedings A and I look forward to receiving your revision. If you have any questions at all, please do not hesitate to get in touch.

Best wishes
Raminder Shergill
proceedingsa@royalsociety.org
Proceedings A

on behalf of
Professor Chris Garrett
Reviews Editor
Proceedings A

Reviewer(s)' Comments to Author:

Referee: 1

Comments to the Author(s)

This is a comprehensive review of the impacts of ocean acidification on different phytoplankton and macrophytes and their production (and by inference emission) of trace gases important for climate and atmospheric chemistry. This is a young field, and the review shows that few conclusive statements could be made based on the paucity of focused studies, and the diverse designs of the experimental studies. The review concludes with suggestions for remedying the situation.

Major comments

1. To my surprise, the values of pH are barely mentioned in this review about ocean acidification. It would be very helpful to include the pH ranges in the different studies (P6, L18 onwards, and Tables 1 and 2). This would help this reader put the various results in some context.
2. Figure 4 is an important summary figure. The last column “Indirect factors affecting trace gas production” is confusing to this reader. It would be helpful to state, for example, “increased surface stratification and decreased nutrient supply” to get a sense of the indirect factors. It is not clear to me why sea level rise would decrease trace gas production. Also, what are the indirect factors associated with Toxins (last row, last column)?
3. The conclusions of the review seem generic – the system is complicated, we need more studies, longer studies, etc. The manuscript would benefit from a targeted set of recommendations – for example, the pH ranges and durations (what is the lifetime of a phytoplankton/macrophyte generation?) of experiments, the comprehensive suite of observations needed in OA sensitive regions. This reader infers that the target of the experimental studies may be pH of 7.8 predicted for 2100 (P11-L31), as some studies with pH as low as 7.2 (P13-L32) and 6.5 (P15-L10) are deemed beyond the expected range. Yet pH 7.6 and 7.7 have already been measured in the California upwelling system (P20-L8), and natural pH fluctuations over the range of 7.5 to 8.3 have been observed in the Arctic and Southern Oceans (P19-L19).
4. What are the key phytoplankton and macrophyte species for the trace gases under consideration? Do they have different pH thresholds for tolerance/change?
5. The title “Unravelling the impacts...” seems aspirational. The review has not clarified the complicated system for this reader.

Other comments

6. Increases in DIC are not the only cause of acidification. Temperature increases (at constant DIC) also lowers pH.
7. P7-L27 onwards. What about effects on clouds?
8. P17-L16-18. CO is a weak GHG, as stated. However, CO competes with CH₄ for OH, and so high CO in the atm means a longer CH₄ lifetime.
9. P18 L32. 6-10 W/m² of reflected sunlight is the total response (background+perturbation), and not the forcing (perturbation since 1800’s). A reader may confuse this with discussion of radiative forcing in the section on aerosols. It would be better to show the forcing from the modeling studies (P19-L3-4).
10. P21-L19: Please give numbers in the text: decreases in pH and DMS flux, so that -0.041C/TgS could be put in context.
11. Additional references:
 - a. pH measurements at Hawaii Time Series
 - b. US Northeast coast - Rheuban et al. JGR Oceans 2019.
 - c. Osborne EB et al. Nature Geoscience Dec 2019 – California current ecosystems

Referee: 2

Comments to the Author(s)

This manuscript is a review of how ocean acidification influences trace gas cycling and exchange. The authors outline the types of experiments to date and the major findings, covering a variety of trace gases. Their focus, however, is on dimethylsulfide (DMS), as it is the most widely studied

biogenic trace gas in relation to ocean acidification. The authors also give recommendations on conducting future studies. Such a review is an important contribution to our understanding of trace gas cycling in a changing environment and useful for the greater community, as many of the findings to date can be confusing to interpret. The authors are clearly experts in the field and have been comprehensive in their treatment of the topic. I recommend that the review be published after the minor comments below are addressed.

Specific comments:

The manuscript repeatedly discusses biogenic trace gas production, but neglects loss processes almost entirely. I think there should be a bit of text addressing how oceanic loss processes are also influenced by ocean acidification.

Page 8, line 10 – Typo, missing word in “...with blooms of this species associated with the release vast quantities of DMS...”

Page 9, lines 9-15 – Why is DMS the most widely studied compound? Maybe it is worth it to say a few words about this?

Pages 9 and 10 – There is no information in the description of DMS/P/O cycling about cell lysis from death, grazers etc that promotes DMS formation, but on page 10, line 13 there is a reference to grazers. It would be good to include something in the text earlier to give context to this point.

Page 11, line 3 – Typo, too many words in “...hypothesis may not to be universally...”.

Page 12-13, section Reconciling differences within and between experimental techniques - Is there a take home message to this section?

Page 16, lines 10-12 - Missing reference to Marandino et al. (2005) and Schlundt et al. (2017) Schlundt, C., Tegtmeier, S., Lennartz, S. T., Bracher, A., Cheah, W., Krüger, K., Quack, B. and Marandino, C. A. (2017) Oxygenated volatile organic carbon in the western Pacific convective centre: ocean cycling, air–sea gas exchange and atmospheric transport *Atmospheric Chemistry and Physics*, 17. pp. 10837-10854. DOI: 10.5194/acp-17-10837-2017.

Marandino, C. A., de Bruyn, W. J., Miller, S. D., Prather, M. J., and Saltzman, E. S. (2005) Oceanic uptake and the global atmospheric acetone budget *Geophysical Research Letters*, 32 (15). DOI 10.1029/2005GL023285.

Page 18, lines 24-25 – Is this negative feedback from the combined effect of greenhouse warming and ozone depletion in stratosphere?

Page 19, section on CO and COS – The role of CDOM in COS ocean cycling is completely ignored and this should be included as CDOM is the main oceanic precursor.

Page 22, lines 2-5 – missing reference to Arevalo et al. (2015)

Arevalo-Martínez, D. L. , Kock, A. , Löscher, C., Schmitz, R. A. und Bange, H. W. (2015) Massive nitrous oxide emissions from the tropical South Pacific Ocean. *Nature Geoscience*, 8 (7). pp. 530-533. DOI 10.1038/ngeo2469.

Page 22, line 15 – missing second parenthesis after citation

Page 22, section Ocean acidification, warming, and deoxygenation: the multi-stressor effects on marine trace gases– Should there be some mention of atmospheric deposition (in the context of reduced mixing)?

Page 27, line 2 - missing second parenthesis after citation

Figure 1 – COS and CS₂ have large net ocean sources that are not included in the schematic (can be included in the DOM area).

Figure 2 – Is there any relationship between DMS (initial) concentrations and percent changes? The amounts in panel f are huge. Is this correct? I did not check the original citation.

Figure 4 – Why is uv-r considered local? In the chemical titles – what exactly is the title of the short lived box; isoprene is one HC; where is COS? In terrestrial environments (also in the ocean), there is a positive correlation of isoprene with temperature, but this is not included here. What is the difference between blank boxes and question marks? Are microbes and phytoplankton used interchangeably (ignoring bacteria)? Because bacterial metabolism may increase with increasing temperature, which may have the effect of increasing DMS, which is not included in the table.

Table 2 – Isoprene and CS₂ are not included anywhere.

Supplementary material

Table S1 – perhaps add information to convey season, location (e.g. temperature)

Table S2 – add season

Table S3 – typo, make CO₂ subscript; maybe a mistake last line, as text says methyl iodide in Bergen experiment, but here states CH₂I₂ in Svalbard.

Author's Response to Decision Letter for (RSPA-2019-0769.R0)

See Appendix A.

RSPA-2019-0769.R1 (Revision)

Review form: Referee 1

Is the manuscript an original and important contribution to its field?

Excellent

Is the paper of sufficient general interest?

Excellent

Is the overall quality of the paper suitable?

Excellent

Can the paper be shortened without overall detriment to the main message?

Yes

Do you think some of the material would be more appropriate as an electronic appendix?

No

Do you have any ethical concerns with this paper?

No

Recommendation?

Accept as is

Comments to the Author(s)

The authors have addressed satisfactorily the issues raised in my first review. I recommend the ms be published.

Review form: Referee 2

Is the manuscript an original and important contribution to its field?

Excellent

Is the paper of sufficient general interest?

Excellent

Is the overall quality of the paper suitable?

Excellent

Can the paper be shortened without overall detriment to the main message?

Yes

Do you think some of the material would be more appropriate as an electronic appendix?

No

Do you have any ethical concerns with this paper?

No

Recommendation?

Accept as is

Comments to the Author(s)

The authors have adequately addressed my comments and those of the other reviewer, in my opinion. The paper should be published.

Decision letter (RSPA-2019-0769.R1)

10-Mar-2020

Dear Dr Hopkins

On behalf of the Reviews Editor, I am pleased to inform you that your manuscript entitled "The impacts of ocean acidification on marine trace gases and the implications for atmospheric chemistry and climate" has been accepted in its final form for publication in Proceedings A.

Our Production Office will be in contact with you in due course. You can expect to receive a proof of your article soon. Please contact the office to let us know if you are likely to be away from e-mail in the near future. If you do not notify us and comments are not received within 5 days of sending the proof, we may publish the paper as it stands.

Your article has been estimated as being 35 pages long. Our Production Office will inform you of the exact length at the proof stage.

Under the terms of our licence to publish you may post the author generated postprint (ie. your accepted version not the final typeset version) of your manuscript at any time and this can be made freely available. Postprints can be deposited on a personal or institutional website, or a recognised server/repository. Please note however, that the reporting of postprints is subject to a media embargo, and that the status the manuscript should be made clear. Upon publication of the definitive version on the publisher's site, full details and a link should be added.

You can cite the article in advance of publication using its DOI. The DOI will take the form: 10.1098/rspa.XXXX.YYYY, where XXXX and YYYY are the last 8 digits of your manuscript number (eg. if your manuscript number is RSPA-2017-1234 the DOI would be 10.1098/rspa.2017.1234).

For tips on promoting your accepted paper see our blog post:

<https://blogs.royalsociety.org/publishing/promoting-your-latest-paper-and-tracking-your-results/>

Thank you for your submission. On behalf of the Editors of the journal, we look forward to your continued contributions to the Journal.

Best wishes
Raminder Shergill,
Proceedings A Editorial Office
proceedingsa@royalsociety.org

on behalf of
Professor Chris Garrett
Reviews Editor
Proceedings A

Reviewer(s)' Comments to Author:

Referee: 2

Comments to the Author(s)
The authors have adequately addressed my comments and those of the other reviewer, in my opinion. The paper should be published.

Referee: 1

Comments to the Author(s)
The authors have addressed satisfactorily the issues raised in my first review. I recommend the ms be published.

Appendix A

Response to Referees for Manuscript ID RSPA-2019-0769:

“Unravelling the impacts of ocean acidification on marine trace gases and the implications for atmospheric chemistry and climate”

by Frances E. Hopkins, Parvatha Suntharalingam, Marion Gehlen, et al.

We are grateful to the referees for their positive view of our manuscript and their thorough assessments which will bring great improvements. The referees comments are shown *in italics*, with our responses shown **in bold**. Page and line numbers in our response refer to the revised version.

1. Response to Referee: 1

1.1 This is a comprehensive review of the impacts of ocean acidification on different phytoplankton and macrophytes and their production (and by inference emission) of trace gases important for climate and atmospheric chemistry. This is a young field, and the review shows that few conclusive statements could be made based on the paucity of focused studies, and the diverse designs of the experimental studies. The review concludes with suggestions for remedying the situation.

We thank the reviewer for recognising the comprehensiveness and importance of our review, and we have taken all of their comments into consideration to bring improvements to the manuscript.

1.2 To my surprise, the values of pH are barely mentioned in this review about ocean acidification. It would be very helpful to include the pH ranges in the different studies (P6, L18 onwards, and Tables 1 and 2). This would help this reader put the various results in some context.

We have made changes to Table 1 to incorporate the CO₂ and pH treatments used for each experiment to provide this important information to the reader. Additionally we have altered the text in the section ‘Experimental evidence: exploring effects of OA on marine trace gases’. This section now includes discussion of the pH ranges/treatments used in the different studies and now reads as follows (new text underlined):

P6, L183 onwards: “Our knowledge of the effects of OA on marine trace gas production stems from the results of a suite of experimental approaches, summarized in Table 1. At the simplest level, on an experimental spectrum of complexity, are incubations with single species algal cultures (<1 L, 2 – 3 replicates, and 7 – 40 d). This is an approach which, given the reduced complexity, serves as a means to establish baseline concepts and identify the most sensitive or relevant physiological processes and mechanisms for trace gas production. Studies have considered ambient CO₂ vs one high CO₂ treatment: 370 - 395 μatm compared to 750 – 1000 μatm, corresponding to pH treatments of 8.1 – 8.3 (ambient) and 8.0 – 7.7 (High CO₂) (see Table 1). Of greater complexity and closer to actual ocean conditions are *in situ* mesocosm experiments, essentially giant ‘test tubes’ that allow large-scale, field-based, community-level assessments of the effects of OA on natural surface ocean communities (2400 – 75000 L, 1 – 3 replicates, and 25 – 35 d). Mesocosms provide an understanding of the net effects on the whole community response to OA, in many cases investigated under conditions of high productivity and growth associated with a phytoplankton bloom. Earlier experiments considered two triplicated CO₂ treatments (‘Ambient’ CO₂ 300 – 400 μatm, pH 8.2 – 8.1 vs High CO₂ 700 – 900 μatm, pH 7.9 – 7.8) (Vogt et al. 2008, Hopkins et al. 2010, Kim et al. 2010, Avgoustidi et al. 2012) (see Table 1). Later experiments considered a wider range of CO₂/pH treatments (175 – 3000 μatm, pH 8.3 – 7.3) using a gradient of treatments levels across up to nine mesocosm enclosures, and allowing linear relationships between

CO₂/pH and response parameters to be determined (Archer et al. 2013, Hopkins et al. 2013, Park et al. 2014, Webb et al. 2015, Webb et al. 2016, Archer et al. 2018) (see Table 1). The observed OA effects on trace gases may reflect a combination of stress responses including acclimation, population re-structuring and associated adaptation (Bach et al. 2016). Shipboard microcosm experiments are a useful tool to bridge the gap between complex mesocosm experiments and simple culture experiments (5 – 10 L, 3 – 12 replicates, and 4 – 10 d). Conducting multiple short-term experiments over extensive spatial scales enables both the physiological effects of OA to be assessed, as well as the spatial variability in responses of surface ocean communities to future OA scenarios. The experimental design, involving relatively small incubation volumes of 5 – 10 L, allows multiple CO₂ treatments to be considered. For 18 experiments performed over a range of temperate and polar waters, Hopkins and Archer (2014) and Hopkins et al. (2020) used 4 CO₂ treatments in triplicate (Mid: 533.4 ± 40.0 μatm, pH 7.9 ± 0.03, High: 673.8 ± 82.2 μatm, pH 7.8 ± 0.1, High+: 841.5 ± 128.2 μatm, pH 7.8 ± 0.1, High++: 1484.0 ± 104.0 μatm, pH 7.5) and an ambient control (320.2 ± 38.3, pH 8.1 ± 0.1). Hussherr et al. (2017) adopted a different approach by exposing phytoplankton communities within 6 single incubations to a pH gradient, from 509 μatm (pH 7.94) to 3296 μatm (pH 7.16) (see Table 1). In the following section, we use information from all three types of experiments to consider the impacts of OA on trace gas production from the cellular to the community level, and in Table 2 we provide an overview of the types of response to OA that may result in changes in trace gas production”.

1.3 Figure 4 is an important summary figure. The last column “Indirect factors affecting trace gas production” is confusing to this reader. It would be helpful to state, for example, “increased surface stratification and decreased nutrient supply” to get a sense of the indirect factors. It is not clear to me why sea level rise would decrease trace gas production.

We have taken the referee’s comments on board, and those of referee 2, and we have made a number of significant changes to the table in Figure 4 to improve the clarity of the information presented in the table. This new version can be found at the end of this document. We have removed reference to ‘sea level rise’.

Also, what are the indirect factors associated with Toxins (last row, last column)?

We have removed the column referring to ‘Toxins’ as it seemed to be creating confusion and was thus deemed unnecessary.

1.4 The conclusions of the review seem generic – the system is complicated, we need more studies, longer studies, etc. The manuscript would benefit from a targeted set of recommendations – for example, the pH ranges and durations (what is the lifetime of a phytoplankton/macrophyte generation?) of experiments, the comprehensive suite of observations needed in OA sensitive regions. This reader infers that the target of the experimental studies may be pH of 7.8 predicted for 2100 (P11-L31), as some studies with pH as low as 7.2 (P13-L32) and 6.5 (P15-L10) are deemed beyond the expected range. Yet pH 7.6 and 7.7 have already been measured in the California upwelling system (P20-L8), and natural pH fluctuations over the range of 7.5 to 8.3 have been observed in the Arctic and Southern Oceans (P19-L19).

We have rearranged some text, and added some further information to the text, to take experimental duration etc. into consideration (underlined):

P26, L836 onwards: “Of course, understanding the biological mechanisms (and their regulation) will be crucial for interpreting the trace gases response to OA, using both model organisms in the laboratory and natural communities within field experiments, and addressing

the current shortcomings requires an improved experimental approach. Although short-term OA studies provide useful information on the physiological plasticity of surface ocean communities and associated trace gas production, and existing levels of adaptation to fluctuations in carbonate chemistry, such experiments cannot accommodate the potential for evolutionary adaptation of planktonic communities (e.g. Lohbeck et al. 2012; Lohbeck et al. 2014; Pančić et al. 2015; Wang et al. 2016). Therefore, it would be beneficial to carry out longer-term experimental studies, encompassing multiple generations, in order to detect adaptation of planktonic communities to OA and other climate change stressors. Such adaptation in phytoplankton becomes evident after only a few hundred generations, representative of a period of ~6 -12 months (e.g. Lohbeck et al. 2012). Parallel measurements of process rates and standing stocks of trace gases would provide greater insight into the role of OA in influencing trace gas production. However, the implementation of long-term experiments of this kind are likely to be limited to culture conditions using isolated strains, and thus at the expense of other important ecological and biogeochemical interactions (see Gattuso and Riebesell 2014). Ecological level experiments will still involve a trade-off in terms of duration and number of generations, but will continue to provide important information on the role of species interactions and succession on trace gas production. Both experimental approaches could integrate multiple stressors, thus closing some of the gaps in our understanding of the trace gas response to climate change”.

Although a targeted set of recommendations could be of use, we feel that this level of detail is outside the scope of this paper. Specific literature on the design of OA/multistressor experiments is already available so we have referred the reader to the appropriate key papers where such information has previously been reviewed and presented:

P27, L872: “Such studies would be complex in design and implementation, and require community-wide collaborative efforts, involving researchers from multiple disciplines and significant levels of financial investment (see Riebesell et al. 2011; Riebesell and Gattuso 2014; Boyd et al. 2018)”.

Riebesell, U., Fabry, V.J., Hansson, L., Gattuso, J.-P. (2011) Guide to best practices for ocean acidification research and data reporting. Office for Official Publications of the European Communities, Luxembourg, 258 pp. doi: [10.2777/66906](https://doi.org/10.2777/66906).

Riebesell, U. and Gattuso, J.P. (2014). Lessons learned from ocean acidification research. Nature Climate Change, **5**, (1), 12.

Boyd, P. W., S. Collins, S. Dupont, K. Fabricius, J. P. Gattuso, J. Havenhand, D. A. Hutchins, U. Riebesell, M. S. Rintoul, M. Vichi, H. Biswas, A. Ciotti, K. Gao, M. Gehlen, C.L. Hurd, H. Kurihara, C.M. McGraw, J.M. Navarro, G.E. Nilsson, U. Passow, H.-O. Pörtner (2018). "Experimental strategies to assess the biological ramifications of multiple drivers of global ocean change—A review." Global Change Biology **24**(6): 2239-2261.

1.5 What are the key phytoplankton and macrophyte species for the trace gases under consideration? Do they have different pH thresholds for tolerance/change?

A great question but we don't currently have enough information to answer this. The majority of studies consider natural mixed communities (mesocosms and shipboard microcosms), such that the resultant net production of trace gases is the result of complex, interacting production and consumption pathways that cannot be attributed to any one species.

For the single species culture work that has been done, there has been somewhat of a focus on the coccolithophore *Emiliania huxleyi*. However, there is much debate about the response of coccolithophores to high CO₂, with large discrepancies in results reported between species and within strains of the same species (see Liu et al. 2018 and references therein). Even less is known about how this variability in response may translate into trace gas production and this kind of discussion and speculation is probably outside the scope of this review.

Liu, Yi-Wei, Robert A. Eagle, Sarah M. Aciego, Rosaleen E. Gilmore, and Justin B. Ries. "A coastal coccolithophore maintains pH homeostasis and switches carbon sources in response to ocean acidification." *Nature communications* 9, no. 1 (2018): 1-12.

1.6 The title "Unravelling the impacts..." seems aspirational. The review has not clarified the complicated system for this reader.

Having taken the referee's view into consideration, we agree that the original title was perhaps somewhat aspirational and we have decided to remove 'Unravelling...'. Title now reads:

The impacts of ocean acidification on marine trace gases and the implications for atmospheric chemistry and climate

1.7 Increases in DIC are not the only cause of acidification. Temperature increases (at constant DIC) also lowers pH.

Ocean acidification can indeed be directly and instantaneously regulated by seawater temperature at a fundamental chemical level (Humphreys 2017), simply due to the re-equilibration of the carbonate chemistry reactions in seawater. Assuming no CO₂ exchange, an increase in temperature results in decreased pH and [HCO₃⁻] and increased [CO₂(aq)] and [CO₃²⁻]. However, it is unclear how influential this process is relative to CO₂ uptake in the surface oceans given that the assumption of no CO₂ exchange would not be relevant. We do not feel that it is necessary to include discussion of this in our review.

Humphreys, M.P., 2017. Climate sensitivity and the rate of ocean acidification: future impacts, and implications for experimental design. *ICES Journal of Marine Science*, 74(4), pp.934-940.

1.8 P7-L27 onwards. What about effects on clouds?

We thank the referee for pointing out this oversight. We have added the following text and accompanying references:

P8, L254: "DMS is also a major source of cloud condensation nuclei (CCN) via its rapid gas phase oxidation to sulfuric acid (H₂SO₄), which influences the radiative properties of clouds, both microscopically via cloud droplet number concentration and effective radius, and at the larger scale by influencing cloud abundance, albedo and lifetime (Brooks and Thornton 2018, Galí et al. 2018, Sanchez et al. 2018)".

1.9 P17-L16-18. CO is a weak GHG, as stated. However, CO competes with CH₄ for OH, and so high CO in the atm means a longer CH₄ lifetime.

This sentence has been rewritten to include this information:

P18, L569: "In the atmosphere, CO is a greenhouse gas with a radiative forcing effect of ~2x CO₂ (on a molecule for molecule basis). Furthermore, CO indirectly affects the climate by

out-competing CH₄ in reactions with tropospheric OH radicals, resulting in enhanced concentrations of this far more potent greenhouse gas (CH₄ ~25x CO₂, see following section Methane) (Forster et al. 2007)”

1.10 P18 L32. 6-10 W/m² of reflected sunlight is the total response (background+perturbation), and not the forcing (perturbation since 1800's). A reader may confuse this with discussion of radiative forcing in the section on aerosols. It would be better to show the forcing from the modeling studies (P19-L3-4).

We have added additional information to this section (underlined). Now reads:

P20, L624: “The modelling studies by Six et al. (2013) and Schwinger et al. (2017) indeed show a significant radiative forcing and surface temperature increase due to OA induced reductions in polar DMS production under high emissions pathways (e.g., a 0.86 W m⁻² reduction in reflected short wave radiation south of 40°S in the study of Schwinger et al)”.

1.11 P21-L19: Please give numbers in the text: decreases in pH and DMS flux, so that -0.041C/TgS could be put in context.

We have now added the requested information into this section (underlined) and it now reads:

P23, L713: “Schwinger et al. (2017) used a fully interactive model, able to simulate a range of feedbacks: they found a global linear relationship between pH-induced changes of DMS sea-air fluxes and a transient surface temperature change of -0.041°C TgS⁻¹ yr⁻¹, driven by reductions in global DMS emissions. These model experiments were conducted with a high emission scenario (RCP8.5) as a baseline, leading to average surface pH reductions of 0.44 and 0.73 units in 2100 and 2200, respectively. The corresponding reduction of DMS fluxes (assuming the “medium” pH-sensitivity of DMS-production of Six et al. 2013) is 4 Tg S yr⁻¹ (17%) in 2100 and 7.3 Tg S yr⁻¹ (31%) in 2200. The simulated additional surface warming has a north-south gradient with much stronger surface warming in the southern hemisphere due to the larger area covered by ocean (Figure 3)”.

1.12 Additional references:

a. pH measurements at Hawaii Time Series

We have added reference to the HOT data, and other sustained ocean observations of pH, P4, L12:

P4, L107: “Globally, a decrease in surface ocean pH of ~0.1 units has already occurred relative to pre-industrial times, with a projected fall of a further ~0.3 units by 2100 under high-emission scenarios (Bopp et al. 2013; Ciais et al. 2014; Gattuso et al. 2015). Sustained ocean observations from seven globally-distributed time series stations, including the northerly Iceland and Irminger Seas, the subtropical Bermuda Atlantic Time-series Study (BATS) and the tropical Hawaiian Ocean Time-series (HOT) show a 0.013 – 0.025 pH unit per decade decline since the 1980s (Bates et al. 2014)”.

Bates, N.R., Y.M. Astor, M. J. Church, K. Currie, J. E. Dore, M. González-Dávila, L. Lorenzoni, F. Muller-Karger, J. Olafsson, and J. M. Santana-Casiano (2014). "A Time-Series View of Changing Surface Ocean Chemistry Due to Ocean Uptake of Anthropogenic CO₂ and Ocean Acidification." *Oceanography* 27, 1: 126-141.

b. US Northeast coast - Rheuban et al. JGR Oceans 2019.

This reference has been added P22, L694:

“Thus CO₂-driven changes in seawater carbonate chemistry occur simultaneously with warming, deoxygenation, localised freshening of the ocean and changes to nutrient dynamics (Gruber 2011; Rheuban et al. 2020)”.

c. Osborne EB et al. Nature Geoscience Dec 2019 – California current ecosystems

This reference has been added at P21, L653:

“Cold DIC-rich subsurface waters are upwelled to the surface layer by trade wind forcing at seasonal and interannual timescales, lowering the pH of surface waters relative to open ocean surface waters (Feely et al. 2008). This enhances the rates of OA within such systems relative to the global surface ocean. Recent data from the California Upwelling System, using a proxy record of fossil foraminifera calcification response, has shown a 35% decrease in [CO₃²⁻] and a drop in pH of 0.21 units since pre-industrial times, a decline of a factor of two greater than the global average (Osborne et al. 2020)”.

2. Response to Referee: 2

2.1 This manuscript is a review of how ocean acidification influences trace gas cycling and exchange. The authors outline the types of experiments to date and the major findings, covering a variety of trace gases. Their focus, however, is on dimethylsulfide (DMS), as it is the most widely studied biogenic trace gas in relation to ocean acidification. The authors also give recommendations on conducting future studies. Such a review is an important contribution to our understanding of trace gas cycling in a changing environment and useful for the greater community, as many of the findings to date can be confusing to interpret. The authors are clearly experts in the field and have been comprehensive in their treatment of the topic. I recommend that the review be published after the minor comments below are addressed.

We thank Referee 2 for their positive overview of our review. We agree that this review is very timely and will provide an essential overview of this topic that will be of interest to the wider ocean acidification (OA) community as well as the wider marine and atmospheric biogeochemistry community, all of whom would benefit from understanding to role of OA in altering the surface ocean production and sea-to-air flux of trace gases.

2.2 The manuscript repeatedly discusses biogenic trace gas production, but neglects loss processes almost entirely. I think there should be a bit of text addressing how oceanic loss processes are also influenced by ocean acidification.

There is currently no available data that specifically addresses the effects of OA on loss processes. When we talk about ‘production’ of trace gases, we mean ‘net production’, i.e. the outcome of the response of both loss and production processes. We have now made this clearer at several points throughout the manuscript:

P5, L135 now reads: “Given the known and predicted effects of OA on biological processes (Riebesell and Gattuso 2015), it is likely that the net production of biogenic trace gases (including both production and loss processes) may be influenced by OA”.

P5, L155 now reads: “Such changes have the potential to indirectly influence trace gas levels by altering the availability of precursors in the dissolved organic carbon pool, and by influencing the rate of bacterial processes that both produce and consume trace gases, and that ultimately result in their net production”.

In the section on *Polar Oceans* section (P20, L629 onwards) within *Cold and naturally carbonated: trace gas emissions from OA sensitive regions*, the potential for loss processes to be affected by OA is mentioned, and we have modified the text (underlined) to make this clearer to the reader:

“A small number of experimental studies report the effects of OA on DMS in polar waters. In a mesocosm experiment in Kongsfjorden, Svalbard Archipelago (78°N) a 35% decrease in DMS at 750 μatm was attributed to a decrease in bacterial DMSP-to-DMS yields (Archer et al. 2013). Similarly, in a microcosm experiment conducted in Baffin Bay, Canadian Archipelago (71°N), a 25% decrease in DMS at ca. 1500 μatm was found, attributed to an OA-related enhancement of DMS loss processes, driven by increases in sulfur demand by the bacterial assemblage (Hussherr et al. 2017)”. These limited results suggest that the net production of DMS during the productive summer season in the Arctic could decrease via bacterioplankton-mediated processes with ongoing OA”.

We have altered text to now read ‘net production’ at P26, L817 and L821.

“The changes in net production of some trace gases such as DMS and N_2O , indicated in OA studies and models,…”

“...but we can have greater certainty that there is the potential for impacts on net production, and so chemistry and climate, with global scale effects”.

P26, L831 now reads:

“Inconsistencies in the effect of OA on trace gas production result from the complexity of trace gas cycling, with involvement of multiple production and loss processes (e.g., phytoplankton species composition, bacterial processes, and grazing activities)”.

2.3 Page 8, line 10 – Typo, missing word in “...with blooms of this species associated with the release vast quantities of DMS...”

Missing ‘of’ has been added to sentence.

2.4 Page 9, lines 9-15 – Why is DMS the most widely studied compound? Maybe it is worth it to say a few words about this?

As we state on P8, L226-227, we focus the discussion on DMS because “the amount of information available for this trace gas currently dwarfs the available information for all others”. There may also be the question of why DMS is so well studied, in general, compared to other trace gases. DMS has formed the focus of marine biogeochemical surface ocean-lower atmosphere studies for the last 3 – 4 decades due to its ubiquity in the surface oceans, the significant role it plays in the transfer of sulfur from the oceans to the atmosphere and land, and its importance in atmospheric and climate-related processes. To this end, we have added the following underlined text at P8, L226:

“A large proportion of the following discussion focuses on DMS, since the amount of information available for this trace gas currently dwarfs the available information for all others. Furthermore, research into the net production of DMS in the surface oceans has been

prominent within the fields of marine biogeochemistry and sea-air interactions for more than three decades due to the global significance of its role in climatic and atmospheric processes. The significance of oceanic DMS production and emission and its potential role in influencing global climate and atmospheric chemistry was highlighted in the 1980s by the seminal publication of Charlson et al. (1987); this spurred more than three decades of intensive investigation into the marine biogeochemistry, air-sea interactions and climate impacts of oceanic DMS”.

2.5 Pages 9 and 10 – There is no information in the description of DMS/P/O cycling about cell lysis from death, grazers etc that promotes DMS formation, but on page 10, line 13 there is a reference to grazers. It would be good to include something in the text earlier to give context to this point.

We have added a sentence at P8, L238 to address this (underlined):

“DMS is produced via enzymatic breakdown of the algal and bacterial secondary metabolite DMSP (Stefels et al. 2007; Curson et al. 2018). The release of intracellular DMSP into the surrounding seawater, and its subsequent and rapid conversion to DMS, is triggered by a number of processes including the active exudation of DMSP from living cells, and cell lysis during senescence, viral attack or grazing (Stefels et al. 2007)”.

2.6 Page 11, line 3 – Typo, too many words in “...hypothesis may not be universally...”.

The unnecessary ‘to’ has been deleted from this sentence.

2.7 Page 12-13, section Reconciling differences within and between experimental techniques - Is there a take home message to this section?

The referee makes a very fair point – upon re-reading this section, it did seem to lack a conclusion/take home message. Thus we have decided to omit this as a standalone section, and instead combine the key points into the main discussion of results for DMS/DMSP.

We have added the following paragraph at P11, L341, following on from the general discussion of the response of DMS/DMSP during the various mesocosm experiments, and so in the correct context:

“The variety of ways in which data from the nine large-scale mesocosm experiments has been interpreted also creates a further challenge when attempting to unify the DMS response to OA. Time-integration of DMS concentrations across the full duration of each experiment (Figure 2) is a start towards consistent interpretation but a more in-depth approach is required to fully compare experiments and extract a comprehensive evaluation. As an example, a large significant increase in DMS in relation to increased CO₂ was reported by Kim et al. (2010) for their experiment in Korean waters – strongly in contrast to all other studies, and which was attributed to an OA-induced enhancement of dinoflagellate grazers. Such differences highlight the complex responses that can play out within the bounds of a mesocosm experiment, making a general, broad-brush interpretation of results challenging”.

And this paragraph has been added at the end of the section at P12, L365, to provide some insight into the how the differences in experimental design etc. between mesocosms and microcosms may have resulted in conflicting results, the reasons for this, and the ways in which the data from each experimental technique should be interpreted:

“Finally, it should be recognized that clear discrepancies have arisen in the DMS response to OA between different experimental techniques, which may make interpretation of the overall response challenging. For example, the results of shipboard microcosms contrast strongly with those from mesocosms. However, interpretation of the data can be facilitated by an understanding of the strengths and weaknesses of each technique, and the specific hypotheses each technique is designed to address. Each approach provides valuable information on how OA may influence DMS production in the future. Microcosm experiments are necessarily short term (< 10 d), so the response to OA is considered to reflect the physiological plasticity of the community i.e. how well they are adapted to rapidly changing carbonate chemistry, but may not fully capture the effects of shifts in community composition. In contrast, mesocosm experiments are generally much longer (typically ~30 days), allowing multigenerational OA-induced changes in taxonomy and community structure to affect DMS concentrations. The microcosm approach may aid in the identification of OA-sensitive regions in terms of DMS production. On the other hand, mesocosm experiments provide some information on how community composition shifts in response to OA may affect the processes controlling the cycle of DMSP and DMS and hence determine their concentrations”.

2.8 Page 16, lines 10-12 - Missing reference to Marandino et al. (2005) and Schlundt et al. (2017)
Schlundt, C., Tegtmeier, S., Lennartz, S. T., Bracher, A., Cheah, W., Krüger, K., Quack, B. and Marandino, C. A. (2017) Oxygenated volatile organic carbon in the western Pacific convective centre: ocean cycling, air–sea gas exchange and atmospheric transport *Atmospheric Chemistry and Physics*, 17. pp. 10837-10854. DOI: 10.5194/acp-17-10837-2017.

Marandino, C. A., de Bruyn, W. J., Miller, S. D., Prather, M. J., and Saltzman, E. S. (2005) Oceanic uptake and the global atmospheric acetone budget *Geophysical Research Letters*, 32 (15). DOI 10.1029/2005GL023285.

We agree that it was an oversight to not include these references, so they have now been added to the text in the appropriate place.

2.9 Page 18, lines 24-25 – Is this negative feedback from the combined effect of greenhouse warming and ozone depletion in stratosphere?

The suggestion of a negative feedback on climate is based solely on the decrease in net N₂O production resulting in a reduction in the greenhouse warming potential of this species. For clarity we have altered this sentence, and it now reads (P17, L544):

“Consequently, the evidence to date suggests the influence of OA may have a small negative feedback on climate change via a reduction in radiative forcing attributed to marine N₂O emissions”.

2.10 Page 19, section on CO and COS – The role of CDOM in COS ocean cycling is completely ignored and this should be included as CDOM is the main oceanic precursor.

We have added additional information and references to this section (underlined) and it now reads:

P18, L577: “Carbonyl sulfide (OCS) is the most abundant sulfur-containing trace gas in the atmosphere, with marine emissions contributing significantly to the total global budget

(Lennartz et al. 2019).OCS is produced in the surface ocean via reactions between UV radiation and CDOM (Uher and Andreae 1997; Lennartz et al. 2019)”.

2.11 Page 22, lines 2-5 – missing reference to Arevalo et al. (2015)

Arevalo-Martínez, D. L. , Kock, A. , Löscher, C., Schmitz, R. A. und Bange, H. W. (2015) Massive nitrous oxide emissions from the tropical South Pacific Ocean. *Nature Geoscience*, 8 (7). pp. 530-533. DOI 10.1038/ngeo2469.

Reference has been added.

2.12 Page 22, line 15 – missing second parenthesis after citation

Missing parenthesis has been added.

2.13 Page 22, section *Ocean acidification, warming, and deoxygenation: the multi-stressor effects on marine trace gases– Should there be some mention of atmospheric deposition (in the context of reduced mixing)?*

We assume the referee is referring to the atmospheric deposition of nutrients, a process which may become more significant in systems that become increasingly stratified with climatic change, reducing the recycling/upwelling of nutrients from deep waters. We have added the following text (underlined) to this section to address the referees concern:

P22, L687 onwards: “Warming and freshening enhances surface ocean stratification (Capotondi et al. 2012), which in turn decreases mixed layer depth and reduces the entrainment of nutrients into the euphotic layer, whilst resulting in higher levels of irradiance experienced by organisms (Bopp et al. 2013). This alleviates light limitation at high latitudes, but enhances nutrient limitation at low- to mid-latitudes. Reductions in nutrient entrainment may be compensated for by the atmospheric deposition of anthropogenic aerosols, which itself could be countered by future improvements in air quality standards (Wang et al. 2015). Thus CO₂-driven changes in seawater carbonate chemistry occur simultaneously with warming, deoxygenation, localised freshening of the ocean and changes to nutrient dynamics (Gruber 2011)”.

2.14 Page 27, line 2 - missing second parenthesis after citation

In fact, it seems the first parenthesis was erroneous, so this has been deleted.

2.15 Figure 1 – COS and CS₂ have large net ocean sources that are not included in the schematic (can be included in the DOM area).

OCS and CS₂ have been added to the schematic.

2.16 Figure 2 – Is there any relationship between DMS (initial) concentrations and percent changes? The amounts in panel f are huge. Is this correct? I did not check the original citation.

There is no clear relationship between initial DMS concentrations and the percent change under high CO₂. There are some complicating factors – for example, not all the experiments started DMS measurements on ‘Day 1’ of the experimental period (see Panel a, c, d).

Yes amounts in panel f are correct. The high DMS concentrations during this experiment occurred during the breakdown of the bloom during heavy grazing activity.

2.17 Figure 4

Why is uv-r considered local?

This was to indicate that it wasn’t climate-related, but to make the table clearer, we have now changed the titles to “*Directly influenced by climate*” and “*Locally variable/indirectly influenced by climate*”.

In the chemical titles – what exactly is the title of the short lived box; isoprene is one HC; where is COS?

HC was for ‘halocarbons’. We have changed the title to: **Other trace gases: Halocarbons, CO, OCS, CS₂, isoprene**

In terrestrial environments (also in the ocean), there is a positive correlation of isoprene with temperature, but this is not included here.

We assume the referee is referring to the work of Dani and Loreto 2017, wherein they argue that phytoplankton isoprene producers favour warmer latitudes (as opposed to cold-water favouring DMS emitters). This could imply that as warmer waters extend towards the higher latitudes with climate change, there could be an increase in the geographical extent of isoprene producers in the oceans. To acknowledge this, we have added isoprene with an ‘up’ arrow into the row that summarises the response to warming.

We have also added some additional text to the manuscript to include some discussion of this important point (underlined):

P23, L723 onwards: “Both models described here are parameterized using the empirical relationship between pH and DMS observed in a number of mesocosm studies (Vogt et al. 2008; Hopkins et al. 2010; Avgoustidi et al. 2012; Archer et al. 2013), whilst recognizing that the level of understanding of the DMS response to OA within these experiments is limited. It should also be noted that this data considers OA as a single stressor, with a complete lack of information for other key climate stressors such as ocean warming. Furthermore, our interpretation of the DMS response between mesocosm studies is confounded by inconsistencies in composition and physiological status of starting communities and experimental set up (e.g., volume of seawater, method of acidification, inorganic nutrient additions, inclusion/exclusion of higher trophic levels, light and UV cycles, mixing, wall effects/cleaning) that make it difficult to draw direct comparisons (see discussion below). To increase the accuracy of model outcomes and facilitate a better understanding of the future feedbacks and climatic effects, improved comparison and integration of all DMS data from mesocosm experiments are required. For example, normalizing for differences in experimental design, community structure, and carbonate chemistry dynamics could lead towards a more accurate empirical relationship between pH and DMS. Where the DMS representation in a model is detailed enough, e.g. Polimene et al. (2012), it would be beneficial to include the effects of OA on the processes controlling DMS production in the

surface ocean, for example, using data from short term, small scale experiments, such as shipboard microcosms (e.g. Hopkins and Archer, 2014, Hussherr et al. 2017; Hopkins et al. 2020).

Finally, there is a large gap in our understanding of the response of net DMS production to other climate stressors in community-level experiments, in particular the response to increasing temperatures. Only a single mesocosm to date has considered the combined effects of OA and temperature (Kim et al. 2010). Our understanding of the multi-stressor response of DMS, and other trace gases, would be improved with a greater understanding of such processes. For example, Dani and Loreto (2017) hypothesise that phytoplankton isoprene emitters favour warmer latitudes, as opposed to cold-water favouring DMS emitters. This may imply that as warmer waters extend towards higher latitudes with climate change, there could be an increase in geographical extent of oceanic isoprene producers to the detriment of DMS producers. However, further work is now required to fill such gaps in our understanding”.

Dani, KG Srikanta, and Francesco Loreto. "Trade-off between dimethyl sulfide and isoprene emissions from marine phytoplankton." *Trends in plant science* 22, no. 5 (2017): 361-372.

What is the difference between blank boxes and question marks?

This was an error, the blank boxes should have contained question marks, so these have now been added in.

Are microbes and phytoplankton used interchangeably (ignoring bacteria)?

We have removed the single use of microbial from the table, and now only refer to phytoplankton.

Because bacterial metabolism may increase with increasing temperature, which may have the effect of increasing DMS, which is not included in the table.

An increase in bacterial metabolism could affect many processes that contribute to net DMS production. It may result in changes to bacterial DMSP processing, with a change in the proportion of DMSP lysis vs DMSP catabolism, which could either increase or decrease the DMS yield. There may also be an increase in bacterial DMS consumption, which would result in a decrease in net yield. To take these processes into account we have added ‘processes’ to the table with both an ‘up’ and ‘down’ arrow to show the possible affects.

2.18 Table 2 – Isoprene and CS₂ are not included anywhere.

Isoprene and CS₂ have now been added to the table in the appropriate places.

2.19 Supplementary material

Table S1 – perhaps add information to convey season, location (e.g. temperature)

Table S1 presents the results from unialgal culture experiments performed under laboratory conditions so this information is not relevant.

Table S2 – add season

Month of experiment and season have been added to Table S2.

Table S3 – typo, make CO₂ subscript; maybe a mistake last line, as text says methyl iodide in Bergen experiment, but here states CH₂I₂ in Svalbard.

CO₂ subscript corrected. Yes, should have read CH₃I, not CH₂I₂. This has been corrected.

Additional changes:

Hopkins et al. 2018 has been replaced with Hopkins et al. 2020 as this paper has now been published in Biogeosciences.

Stressor	CO ₂	N ₂ O	CH ₄	DMS	NH ₃	Other trace gases (Halocarbons, CO, OCS, CS ₂ , OVOCs, isoprene)	Indirect factors affecting trace gas production & emission
Directly influenced by climate							
Warming	↓: ↓solubility ↑: ↓ primary production ↑: ↑ respiration ↑: ↑ DOM & TEP	↓: ↓ solubility ↑: ↑nitrification	↓: ↓ solubility ↑: ↑methanogenesis ↑: ↑ hydrate decomposition	↓: ↓ solubility ↑: ↑ nutrient stress ↓: ↓ phytoplankton population shifts ↑/↓: changes to bacterial processes	↓: ↓ solubility	↓: ↓ solubility (all) ↑: ↑ global phytoplankton distribution shifts (isoprene)	↓: ↑ surface stratification & decreased nutrient supply to surface ocean ↓: ↓ C fixation & export ↓: changes to ventilation & thermohaline circulation ↑: ↑ winds & storm intensity ↑: ↑ dust deposition
Ocean Acidification	↑: ↑ DOM & TEP ↔: ↔ primary production	↓: ↓ nitrification	↔: no response	↓: ↓ phytoplankton population shifts ↓: ↓ phytoplankton biomass ↑: ↑ physiological stress ↑/↓: ↓ bacterial processes ↑: ↑ grazing on phytoplankton	↓: ↓ NH ₄ ⁺ (phase shift)	↓: ↓ phytoplankton population shifts (iodocarbons) ↔: no response (biomocarbons) ?: CO, OCS, CS ₂ , OVOCs, isoprene	↑: ↓ primary production & C export ↓: ↓ export production (decreased carbonate production) ↑: ↑ trace metal availability (phase shifts)
Hypoxia	↑: ↑respiration	↑: ↑nitrification ↑/↓: ↓denitrification	↑: ↑ anaerobic microsites ↑: ↑ methanogenesis	↑: ↑ anoxic stress	↑: ↑ammonification	?	↓: ↓ nutrient supply in upwelling regions
Locally variable/indirectly influenced by climate							
Increased UV-R	↓: ↓ respiration ↑: ↑ photoproduction	↓: ↓nitrification	↔	↑: ↑ photooxidative stress ↓: ↓ phytoplankton biomass	?	↑: ↑ photoproduction (CO, halocarbons, isoprene) ↑: ↑ photooxidative stress ↓: ↓ phytoplankton biomass	↓: ↓ phytoplankton biomass
Increased nutrient loading	↑: ↑respiration ↑: ↑ DOC	↑: ↑ nitrification ↑: ↑denitrification	↑: ↑ anaerobic microsites ↑: ↑ methanogenesis	↑: ↑ primary production ↑: ↑ phytoplankton biomass	↑: ↑ DON	↑: ↑ photoproduction (CO, halocarbons, isoprene)	↑: ↑ primary production & C export ↑: ↑ occurrence of HABs ↑: ↑ hypoxia
Increased particle loading	↑: ↑respiration ↑: ↑ DOC	↑: ↑nitrification ↑: ↑denitrification	↑: ↑ anaerobic microsites ↑: ↑ methanogenesis	↓: ↓ primary production	↑: ↑ DON	↓: ↓ decreased photoproduction (CO, halocarbons, isoprene)	↓: ↓ bioturbation ↑: ↑ hypoxia

Figure 4. Summary of our knowledge on multiple stressors and their anticipated direct and indirect effects on trace gas production. Coloured arrows represent known/anticipated trace gas response (red = increase, blue = decrease, green = no net change), and black arrows describe the direction of change of the related process. HABs = Harmful Algal Blooms. TEP = Transparent Exopolymer Particles. DON =Dissolved Organic Nitrogen.